# Room-temperature cavity exciton-polariton condensation in perovskite quantum dots

Ioannis Georgakilas[1,2,6], David Tiede[3,6], Darius Urbonas [1], Rafał Mirek [1], Clara Bujalance[3], Laura Caliò [3], Virginia Oddi[1,4], Rui Tao[4,5], Dmitry N. Dirin [4,5], Gabriele Rainò [4,5], Simon C. Boehme [4,5], Juan F. Galisteo-López[3], Rainer F. Mahrt [1], Maksym V. Kovalenko [4,5] ✉, Hernán Miguez [3] ✉ & Thilo Stöferle [1] ✉

The exploitation of the strong light-matter coupling regime and exciton-polariton condensates has emerged as a compelling approach to introduce strong interactions and nonlinearities into numerous photonic applications. The use of colloidal semiconductor quantum dots with strong three-dimensional confinement as the active material in optical microcavities would be highly advantageous due to their versatile structural and compositional tunability and wet-chemical processability, as well as potentially enhanced, confinement-induced polaritonic interactions. Yet, to date, exciton-polariton condensation in a microcavity has neither been achieved with epitaxial nor with colloidal quantum dots. Here, we demonstrate room-temperature polariton condensation in a thin film of monodisperse, colloidal $CsPbBr_3$ quantum dots, placed in a tunable optical resonator with a Gaussian-shaped deformation serving as wavelength-scale potential well for polaritons. The onset of polariton condensation under pulsed optical excitation is manifested in emission by its characteristic superlinear intensity dependence, reduced linewidth, blueshift, and extended temporal coherence.

Cavity exciton-polaritons are bosonic quasi-particles that are part light and part matter, arising from strong coupling of semiconductor excitons and photonic modes in optical microcavities[1,2]. They have attracted intense attention due to their ability to form non-equilibrium Bose-Einstein condensates, *i.e.*, a macroscopically occupied coherent quantum state[3]. The differences and beneficial aspects of such polariton condensates compared to photon lasers have been thoroughly explored[4–6], and they provide an excellent basis for studying and exploiting quantum fluids of light[7–9] and building optoelectronic devices that benefit from their nonlinearity and interactions[10,11]. Finally reaching the quantum regime[12] by achieving polariton blockade[13]

would enable strongly correlated polariton phases and applications in quantum information processing, but thus far has been notoriously elusive, even with very tight photonic confinement[14,15]. Exploring more strongly confined excitons like those in three-dimensionally (3D) confined nanoscale quantum dots (QDs) appears to be a promising route that already has enabled single-photon switches at cryogenic temperature[16]. Both enhanced Coulomb interaction of excitons and Pauli blocking from the discretized density-of-states are key ingredients for the blockade regime. Towards this aim, colloidal QDs in particular represent an attractive material platform not only due to their precisely controllable composition, size, and shape, but also their

[1]IBM Research Europe – Zurich, Säumerstrasse 4, 8803 Rüschlikon, Switzerland. [2]Institute of Quantum Electronics, Department of Physics, ETH Zürich, Auguste-Piccard-Hof 1, 8093 Zürich, Switzerland. [3]Multifunctional Optical Materials Group, Institute of Materials Science of Seville, Consejo Superior de Investigaciones Científicas – Universidad de Sevilla (CSIC-US), Américo Vespucio 49, Sevilla 41092, Spain. [4]Laboratory of Inorganic Chemistry, Department of Chemistry and Applied Biosciences, ETH Zürich, Vladimir-Prelog-Weg 1-5/10, 8093 Zürich, Switzerland. [5]Laboratory for Thin Films and Photovoltaics, Empa – Swiss Federal Laboratories for Materials Science and Technology, Ueberlandstrasse 129, 8600 Dübendorf, Switzerland. [6]These authors contributed equally: Ioannis Georgakilas, David Tiede. ✉e-mail: mvkovalenko@ethz.ch; h.miguez@csic.es; tof@zurich.ibm.com

facile wet-chemical synthesis and processing, amenable to a potential future scaling-up of this technology. While strong light–matter coupling has been achieved with various kinds of colloidal II-VI semiconductor nanocrystals and microcavity architectures[17–20], polariton condensation[21] has only been accomplished with quantum-well-like nanoplatelets where strong excitonic confinement is realized only in one dimension. Hence, cavity exciton-polariton condensation has remained beyond reach with ensembles of 3D-confined semiconductor QDs, regardless of whether the QDs were grown colloidally or by epitaxial deposition methods. Supposedly, this can be attributed to the significant inhomogeneous spectral broadening characteristic for the strong 3D-confinement regime.

More recently, lead-halide perovskites have emerged as an attractive alternative to traditional semiconductors due to their exceptional optical properties. Colloidal cesium lead halide QDs exhibit wavelength-tunable emission[22] with near-unity photoluminescence quantum yield[23] and small homogeneous broadening[24] even at room temperature. At cryogenic temperature, they attain extraordinarily high oscillator strength[25] and long coherence time[26,27], which has been exploited to generate cooperative, superfluorescent emission[28]. They have been utilized in a multitude of optoelectronic applications[29], such as quantum light sources[30,31], light emitting diodes (LEDs)[32], solar cells[33], and lasers[34,35]. With thin, bulk-like lead-halide perovskite crystals, room-temperature polariton condensation[36] and various demonstrations of quantum-fluid properties[37,38], condensation in arrays[39,40], and topological polariton lasing[41] have been reported. Compared to these macroscopic single crystals, colloidal nanocrystal QDs have the advantages of wavelength tunability[22] and size-dependent, strong exciton-exciton interactions[42,43], highly engineerable and flexible synthesis, deposition and processing, and a discrete, non-continuous density of electronic states – a feature that has allowed conventional semiconductor QDs to become superior laser gain materials over their quantum well counterparts[44] and likely could have far-reaching implications also for the quantum fluid properties of polariton condensates.

However, condensation with perovskite QDs in the strong confinement regime has thus far not been achieved because of the typically poor optical quality of QD films due to high surface roughness and volume scattering, in combination with broadened excitonic transitions owing to size and energy dispersion. In contrast, recent success in synthesizing size- and shape-monodisperse CsPbBr$_3$ perovskite QDs with up to four distinct and narrow excitonic bands and facile control of the surface chemistry[45] allowed the development of metallic resonators embedding non-scattering QD films and the

observation of strong light-matter coupling in perovskite QD solids[46]. Simultaneously, at a temperature of 10 K, a condensate of propagating waveguide polaritons was reported in a superfluorescent CsPbBr$_3$ QD film[47]. Yet, exciton-polariton condensation in a microcavity has not been realized for any QD platform, neither at cryogenic nor ambient conditions.

Here, we demonstrate room-temperature exciton-polariton condensation in a perovskite-QD solid that is embedded in an open-cavity optical resonator comprising a wavelength-scale Gaussian deformation. In contrast to the previously reported observation where waveguide polaritons condense at $T = 10$ K in an excited, propagating state[47], in our work cavity polaritons condense at room temperature in the ground state of the polariton branch, consistent with condensation processes in traditional bosonic platforms. When tuning the length of the microcavity, strong light-matter coupling with a characteristic anticrossing behavior is observed, evidencing the formation of exciton-polaritons. Above a certain excitation-intensity threshold, polariton condensation gives rise to superlinear emission enhancement, spectral narrowing, blue-shifted emission, and extended temporal coherence.

A thin film of highly size- and shape-monodisperse colloidal CsPbBr$_3$ perovskite QDs (size of $6.85 \pm 0.85$ nm, Supplementary Fig. 1) blended with a small quantity of stabilizing and homogenizing polystyrene, prepared as described in Methods, was placed inside a tunable microcavity, as displayed in Fig. 1a. While the film can develop cracks during drying (Supplementary Fig. 2a), the resulting domains are large and of high optical quality with 1 – 2 nm root-mean-square (rms) surface roughness (Supplementary Fig. 2b). The QD film exhibits multiple well-defined, narrow excitonic transitions in absorption (76 meV full width at half maximum; FWHM) and a single narrow emission peak (89 meV FWHM), as can be inferred from the photoluminescence excitation (PLE) and photoluminescence (PL) spectra in Fig. 1b and Supplementary Fig. 3, which were obtained outside the cavity. The exciton spectral width is dominated by a combination of inhomogeneous and homogeneous broadening at room temperature[45], while the actual damping rate of the transition is around ~1 µeV corresponding to the reported 3 – 5 ns lifetime of CsPbBr$_3$ QDs[25,48,49]. The employed open-cavity structure comprises two halves, separately placed on nanopositioning stages for adjusting their relative position, separation and tilt (Fig. 1a). The lower half consists of a DBR with the perovskite QD film, and the top half consists of a DBR with a Gaussian-shaped deformation of ~2 µm FWHM and ~45 nm depth (see Supplementary Fig. 4 and Methods for details). This deformation acts as a potential well for the polaritons, inducing lateral confinement of their

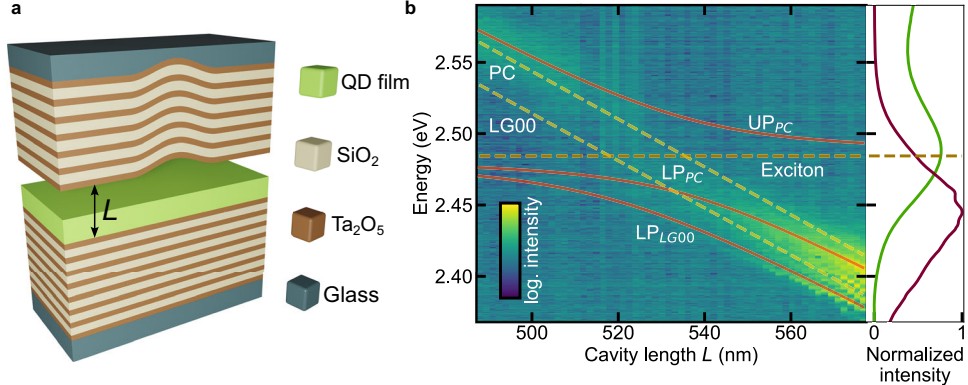

**Fig. 1 | Strong light–matter coupling of CsPbBr$_3$ perovskite QDs in a tunable Gaussian-deformation cavity. a** Sketch of the optical resonator consisting of SiO$_2$/Ta$_2$O$_5$ DBR layer stacks with a Gaussian deformation in the top mirror and a QD film as the emitter layer on the bottom mirror of the cavity with tunable length L. **b** Left panel: White-light transmission spectra as a function of the cavity length. Orange solid lines represent the fits of the upper polariton (UP$_{PC}$) and lower polariton

(LP$_{PC}$) for the PC mode and the lower polariton (LP$_{LG00}$) for the LG00 mode. The horizontal dark yellow dashed line indicates the fitted exciton energy of 2.484 eV, which coincides well with the exciton peak of the PLE measurement. The two dashed bright yellow lines are the purely photonic dispersions of the PC and the LG00 mode. Right panel: PL (purple line) and PLE (green line) spectra of the QD film obtained outside the cavity.

wavefunction and therefore, in addition to the planar cavity (PC) mode, a set of discrete energy states is supported in the form of Laguerre-Gaussian modes[50], denoted as LG$nl$ with the radial quantum number $n$ and azimuthal quantum number $l$. The tight zero-dimensional confinement provided by the Gaussian deformation[51] can lead to lower condensation threshold, due to large spatial overlap of the trapped condensate with the gain (pumped area), and also to enhanced Rabi splitting, thanks to an effective reduction of excitonic and photonic disorder that can lead to energetic inhomogeneity[52,53].

## Results

### Perovskite QD exciton-polariton formation in a zero-dimensional cavity

In contrast to planar cavities, which allow angle-tuning due to their continuous, quasi-parabolic dispersion, the discrete, dispersion-less nature of the localized LG states necessitates the direct measurement of the anti-crossing characteristic in the strong coupling regime via bringing the exciton and the cavity modes into resonance through tuning of the cavity length[1,54]. The sample was illuminated by means of a broadband white-light source with the beam focused on top of the Gaussian deformation. The spectral transmission linewidth measured far from the excitonic resonance is about ~5 meV FWHM, corresponding to a photonic resonator quality factor of $Q \sim 500$. To demonstrate that the system is in the strong light–matter coupling regime, we recorded transmission spectra while tuning the length of the cavity by changing the air gap in situ (Fig. 1b, Supplementary Figs. 5 and 6). The measured spectra are composed of spatially confined LG modes arising from the Gaussian-shaped potential and the PC mode (highest in energy) (Supplementary Fig. 7). By energetically tuning the photonic modes through the exciton resonance, we observe the characteristic anti-crossing behavior of the lower polariton (LP) and upper polariton (UP) branches when entering the strong-coupling regime. We obtain Rabi splitting values of $2\Omega_{PC} = (53 \pm 2)$ meV and $2\Omega_{LG00} = (60 \pm 2)$ meV for the PC mode and the lowest-energy Gaussian mode, respectively, by fitting the data with a coupled-oscillators model. As the air-gap length is not directly measurable in the experiment, the extracted polariton dispersion is compared to a transfer-matrix model simulation to precisely determine the cavity length $L$ (Supplementary Figs. 7 and 8), which we define here as the size of the air gap plus the QD film thickness of 245 nm but excluding the exponential decay length within the DBRs.

### Room-temperature condensation of polaritons in a Gaussian-shaped potential

To drive the system to polariton condensation, we used non-resonant pulsed excitation, with a beam size of 1.5 – 3 μm, similar to the Gaussian deformation's dimensions (see Methods for more details of the setup). Fourier-space imaging was employed to study the angle-resolved emission below and above condensation threshold, as displayed in Fig. 2. Below threshold, for a cavity length of $L \sim 575$ nm, polaritons populate the characteristic states emerging from the dispersion relation of the Gaussian potential, i.e., the LG00 ground state at 2.375 eV and the LG01 first excited state at 2.397 eV (Fig. 2a), corresponding to LG00 exciton-photon detuning of −100 meV and a detuning of −72 meV for the PC mode. Above condensation threshold, polaritons primarily occupy the LG00 ground state of the Gaussian deformation, concomitant with a spectral narrowing of its emission (Fig. 2b). The condensation regime also becomes evident by monitoring the emission in real space where, below threshold, the PC mode and the various LG modes lead to a broader distribution (Fig. 2c), whereas above threshold, only the LG00 mode is observed (Fig. 2d), indicating a single-mode condensate. As is common for many room-temperature polariton condensates[55], we observe condensation at negative cavity detunings from the exciton, which corresponds to a quite photonic nature of the polaritons with excitonic fractions of 7%-10%. From the

intensity ratio $I_{LG01}/I_{LG00}$ below threshold and assuming Maxwell-Boltzmann distribution $I_{LG01}/I_{LG00} = \exp(-(E_{LG01}-E_{LG00}) / k_B T_{eff})$, with $k_B$ being the Boltzmann constant, we obtain an effective polariton temperature $T_{eff} = (228 \pm 10)$ K, which is close to room temperature but indicates not fully complete thermalization, similar to other trapped polariton condensates[56].

To quantify the onset of the condensation process, we monitored the emission spectrum while varying the excitation fluence. Figure 3a shows emission spectra at three different excitation fluences. Below condensation threshold, the LG00 emission spectrum consists of a single peak, attributed to uncondensed polaritons. At the onset of condensation, an additional peak of higher energy and smaller line-width appears, attributed to the emerging condensate. At stronger excitation far above the threshold, the condensate peak dominates the spectrum. Notably, for below-threshold and above-threshold LG00 emission spectra exhibiting a single resolvable peak only, we fitted the spectrum with a single Gaussian function, while near the threshold we performed a fit consisting of the sum of two Gaussians. The integrated counts of the Gaussian fits versus the excitation fluence exhibit a superlinear increase in the emission intensity (Fig. 3b), a signature of the condensation process. Figure 3c shows the behavior of the emission linewidth with increasing fluence for both the uncondensed polaritons peak and the condensate. A drastic reduction of the emission linewidth at $P_{th} = 160$ μJ cm$^{-2}$, which matches the onset of the superlinear emission, indicates the threshold of the condensation process. The observed threshold is comparable to the polariton condensation threshold of 130 μJ cm$^{-2}$ reported in a very similar Gaussian-shaped microcavity with an organic polymer as the active layer[57] but about two orders of magnitude higher than reported for polariton condensates in CsPbBr$_3$ microcrystals[37], consistent with the much smaller effective filling fraction with active perovskite material (due to air gap and ligands) and larger homogeneous and inhomogeneous broadening of our QD solid arising from the increased exciton-phonon coupling and the finite size distribution of QDs, respectively.

A second important signature to consider for the condensation process is the blueshift of the emission peak with increasing excitation fluence, reaching at about 1.7 $P_{th}$ a ~5 meV blueshift with respect to the emission below threshold (Fig. 3d), very similar to literature reports on condensates in CsPbBr$_3$ microcrystals[37]. The origin of the blueshift is currently intensively discussed within the literature, where polariton-polariton interactions[58], excitation-induced changes in the refractive index and saturation effects of the optical transitions are taken into account[59]. The relative contributions of each of the mentioned mechanisms will strongly depend on the exact nature of the semiconductor material; in this respect, it is interesting to evaluate the current case of confined excitons in a QD which may be regarded as an intermediate case between the more explored Wannier-Mott excitons in inorganic semiconductors and the Frenkel excitons in organic semiconductors. On the one hand, in the present system, transient-absorption measurements that allow to track the evolution of various polariton branches, have shown an excitation-density-dependent reduction of the Rabi splitting due to saturation effects, which could be a major contribution to the energy shift[46]. On the other hand, some sample positions exhibited a slightly non-monotonic blueshift with the excitation density near threshold (also when using the non-condensing LG01 state as "reference" to eliminate sensitivity to cavity length drift or jumps), which could arise from polariton-polariton interactions in the dense condensate. In the presented dataset in Fig. 3, the uncondensed polaritons LG00 peak, the condensing LG00 and the non-condensing LG01 experience a similar blueshift, potentially being a result of similar interactions with the exciton reservoir or the onset of saturation that are dependent mainly on the excitation fluence but not the polariton density in the respective state. However, the clearly observed threshold of the blueshift at $P_{th} = 160$ μJ cm$^{-2}$, coinciding with the condensation threshold, might indicate polariton-polariton

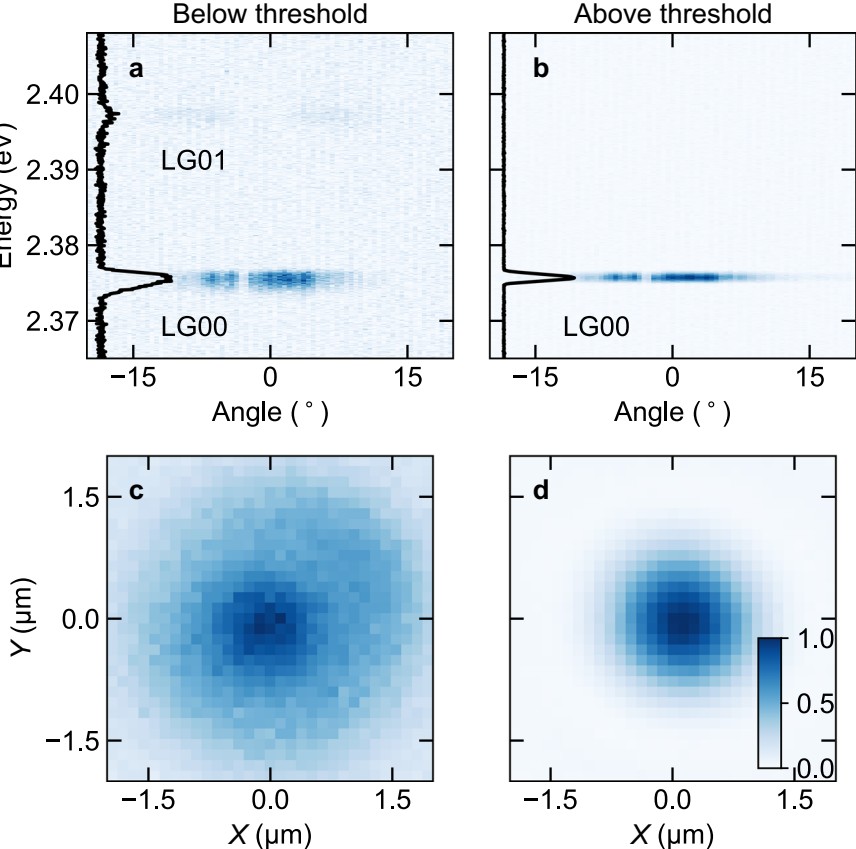

**Fig. 2 | Dispersion and real-space emission of exciton-polariton condensation in a microcavity with a Gaussian deformation. a** Angular dispersion of the emitted light below polariton condensation threshold showing the LG00 ground state at 2.375 eV and, faintly, the LG01 state at 2.397 eV, and **b** above threshold, where polariton condensation in the ground state occurs. Images were acquired in Fourier-space imaging configuration, with the fine substructure (angle-dependent intensity modulation) of the graph being an artifact caused by dust particles inside the spectrograph. The respective angle-integrated spectrum (black solid line) is illustrated on the left side of each panel. **c** Real-space emission acquired using weak continuous-wave excitation, displaying significant background PL, with an emission intensity maximum at the location of the Gaussian deformation, corresponding to a mixture of LG00, LG01 and PC modes. **d** Real-space emission image obtained under strong, pulsed excitation, forming a polariton condensate in the ground state of the Gaussian-shaped potential well, indicating single-mode condensation. The images in (**a**)–(**d**) display normalized emission intensity with the false-color encoding defined by the color bar provided in (**d**).

interactions due to the high polariton density in the condensate regime. Hence, based on the current experimental data, we cannot fully conclude about potential modifications of the polariton interactions due to the excitonic confinement in the QDs. Nevertheless, the blueshift does not saturate and its magnitude stays significantly below both the coupling strength (60 meV) and the energy difference between the LG00 lower polariton and photonic mode (~10 meV), therefore proving that, even for the highest studied excitation fluence, the sample remains in the strong light–matter coupling regime. It is important to note that while the main effects remain quite similar, repeating the excitation-dependent measurements at different material positions and at a second, almost identical, sample led to variations of mode energy, threshold behavior and blueshift, as a result of both film inhomogeneity and small fluctuations of the geometry of the experimental setup (e.g., drift of the cavity length or position) (Supplementary Data Fig. 9). Notably, the material stability was sufficient to obtain multiple measurement runs on the same position without significant, permanent degradation of the perovskite QD solid.

### Coherence measurements below and above condensation threshold

An important characteristic of polariton condensation is extended phase coherence. To probe this condensate feature, we sent the polariton emission through a Michelson interferometer, and interfered the real-space image of the condensate with a centrosymmetric copy of itself on a camera to obtain the spatial and temporal evolution of the first-order coherence. The resulting continuous fringe pattern with constant contrast over the whole emission interferogram (see Fig. 4a) suggests that a single-mode polariton condensate is formed in the LG00 state.

The temporal behavior can be inferred from the decay of the fringe visibility as a function of the delay $\Delta t$ provided by one arm of the interferometer. Below threshold, the coherence decays quickly, with fringes becoming invisible already after a time delay of 0.1 ps (Fig. 4a top panels). Above threshold, conversely, the fringe pattern is still clearly visible at a time delay of 2.8 ps, representing a prolongation of the coherence by more than one order of magnitude (Fig. 4a bottom panels). Figure 4b shows the extracted fringe visibility for different $\Delta t$ values. The experimental data are fitted with a Gaussian autocorrelation function, indicating an above-threshold temporal coherence reaching 5.2 ps FWHM. As with many other polariton systems[60–62], the condensate coherence lasts much longer than the polariton lifetime (here: ~0.65 ps), as in this driven-dissipative system the condensate is continuously replenished through polaritons that relax from the exciton reservoir created by the off-resonant pump pulse (many picoseconds). On top of pump-induced effects, polariton interactions and number fluctuations could be possible explanations[63,64] for the Gaussian temporal decay of the phase coherence.

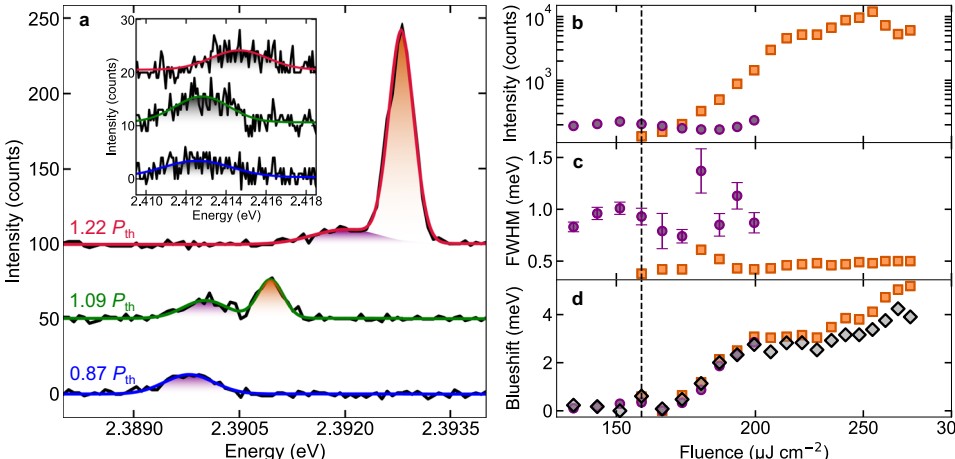

**Fig. 3 | Spectral emission signatures of exciton-polariton condensation.**
**a** Experimental data (black solid lines) and fit (colored solid lines) for three representative emission spectra of the LG00 polariton state, below (0.87 $P_{th}$), near (1.09 $P_{th}$) and above (1.22 $P_{th}$) condensation threshold. The inset shows the simultaneously recorded evolution of the LG01 polariton state, which does not exhibit condensation. The purple, orange and grey shaded areas correspond to the areas defined by the uncondensed LG00, the LG00 condensate and the non-condensing LG01 peaks, respectively. At condensation threshold the PC and LG00 exciton-photon detunings are −57 meV and −85 meV, respectively. **b–d** Excitation-fluence dependence of key signatures obtained from Gaussian fits to the experimental emission spectra where the uncondensed LG00 polariton emission is denoted by purple discs, the LG00 condensate by orange squares, and the non-condensing LG01 emission by grey diamonds: **b** emission intensity, **c** emission linewidth, **d** emission peak energy. Error bars obtained from the fit errors are only significant for the uncondensed LG00 polariton emission linewidth and therefore presented for these data points in (**c**). The condensate peak with its characteristic nonlinear intensity increase and narrow linewidth emerges at a threshold of ~160 μJ cm⁻², indicated by the vertical black dashed line.

## Discussion

In conclusion, we report the observation of room-temperature cavity exciton-polariton condensation in a QD solid, employing a CsPbBr₃ QD film in a tunable DBR microcavity with a wavelength-scale Gaussian-shaped deformation. Polariton formation is inferred from the characteristic anti-crossing between lower and upper polariton branches in cavity-length-dependent transmission measurements and enabled by the existence of extended areas of high optical quality and low surface roughness in the QD film. We demonstrated single-mode room-temperature polariton condensation in the ground state of the Gaussian potential, as evidenced by a threshold for superlinear emission intensity, linewidth narrowing and blueshift, observed both in angle-resolved emission and excitation-fluence-dependent spectral measurements. In the condensate regime, we observed emission in a single, discrete mode – an important aspect for potential device applications – and with an extended temporal coherence reaching 5.2 ps. Notably, with the present excitation beam size and fluence, we have not been able to achieve polariton condensation with planar cavities without Gaussian deformation, which suggests the pivotal role of the strong photonic confinement.

Colloidal perovskite QDs are a promising active material for polaritonic devices due to their unique optical properties as well as their wet-chemical processibility. The here employed methodology for creating a single polariton potential well via a Gaussian-deformation microcavity can be extended to more complex polariton lattices[65], opening a new path for exploiting perovskite QDs in the study of fundamental lattice models towards analog quantum simulations. Moreover, such wavelength-scale potential arrays and patterned excitation beams, or alternatively, resonant excitation schemes[66], can help to determine and observe polariton interactions in the future, and ideally push the system into the polariton blockade regime, like it has been achieved for epitaxially-grown semiconductor QDs in photonic crystal cavities[67].

## Methods

### Stock solution for the synthesis of CsPbBr₃ QDs
Precursor solutions were prepared as reported elsewhere[45]. PbBr₂-TOPO stock solution (0.04 M) was prepared by first dissolving 4 mmol of PbBr₂ (99.999%, Sigma Aldrich) and 20 mmol of tri-n-octylphosphine oxide (TOPO, > 90%, Strem Chemicals) in 20 mL of n-octane (99%, Carl Roth) at 120 °C. The solution was then diluted with 80 mL of hexane (≥ 99%, Sigma Aldrich) and filtered through a 0.2 mm PTFE filter. Similarly, the Cs-DOPA solution (0.02 M) was prepared by mixing 500 mg of Cs₂CO₃ (99.9%, Sigma Aldrich) with 5 mL of diisooctylphosphinic acid (DOPA, 90 %, Sigma Aldrich) in 10 mL of n-octane at 120 °C and subsequent dilution with 135 mL of hexane. The obtained solution was filtered through a 0.2 mm PTFE filter. The 0.13 M lecithin stock solution was prepared by dissolving 2.0 g of lecithin (> 97% from soy, Carl Roth) in 40 mL of hexane, followed by filtering the solution through a 0.2 mm PTFE filter.

### Synthesis of CsPbBr₃ QDs
Colloidal CsPbBr₃ QDs were obtained in accordance with the previously reported synthetic method[45]. PbBr₂-TOPO stock solution (30 mL) was diluted with 180 mL of filtered hexane, followed by the injection of 15 mL of Cs-DOPA stock solution under vigorous stirring. After 4 min of growth, 15 mL of lecithin stock solution were added, and the solution was allowed to stir for 1 min more. The crude solution was concentrated to 15 mL on a rotary evaporator, and 30 mL of acetone acting as an antisolvent were added. QDs were isolated by centrifuging at 20133 g for 1 min and redispersed in 24 mL of dried toluene. QDs were precipitated from this solution by adding 24 mL of dried ethanol and centrifuging the mixture at 20133 g for 1 min. The product was redissolved in 12 mL of toluene, and washing was repeated with 12 mL of ethanol, followed by redissolution in 6 mL of toluene. The last washing with 6 ml of ethanol was performed to remove the excess of lecithin. The obtained QD pellet was redissolved in 2 mL dried toluene to obtain the final 83 mg/mL concentrated CsPbBr₃-QDs dispersion.

### QD-solid preparation and characterization
QD films were prepared following the method described in ref. 46. Chemical reagents and solvents were purchased from Sigma-Aldrich and used without further purification. We first prepared a polystyrene (PS, average $M_w$ ~ 35000) solution at 10 wt% in toluene, stirring at room temperature. Next, we mixed a proportion in volume of

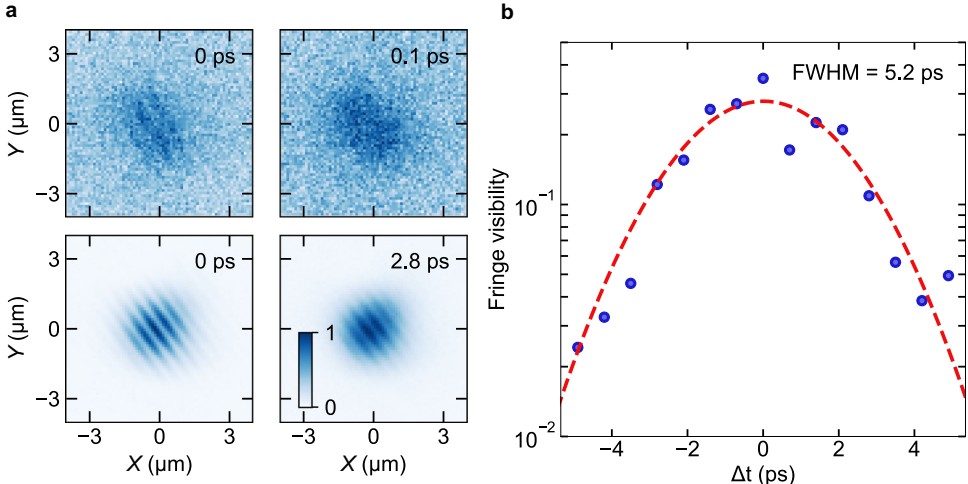

**Fig. 4 | Coherence below and above condensation threshold. a** Real-space interferograms of the emitted signal below (top panels, ~0.2 $P_{th}$, exposure time 10 s) and above (bottom panels, ~2.4 $P_{th}$, exposure time 1.5 s) condensation threshold. While for $\Delta t = 0$ ps interference fringes are observed in both cases, below threshold the fringes have already disappeared after ~0.1 ps, while they are still visible even after ~2.8 ps above threshold. **b** Extracted fringe visibility of the interfered condensate signal at ~2.4 $P_{th}$ excitation fluence for different time delays

$\Delta t$. The fringe visibility at each time delay $\Delta t$ was calculated from the Fourier transform of the images, by dividing the signal at the fringes' spatial period by the total signal of the whole Fourier transformed image. Finally, this fringe visibility is calibrated by comparison with the visibility at zero time delay obtained directly from the fringe minima/maxima in the real space image. Fitting the experimental data with a Gaussian function (red dashed line) shows an extended temporal coherence with 5.2 ps FWHM autocorrelation decay.

1:6.1 of [QDs dispersion at 83 mg/ml]:[PS solution at 96.3 mg/ml], keeping the weight proportion of 84:16 QDs:PS in the film. PS was added to the QD dispersion to improve the stability of the final film. Films used for the optical characterization were prepared on quartz substrates (cleaned by ultrasonic bath with 2% Hellmanex, acetone and 2-propanol) and by spin coating the precursor solution at speeds ranging from 3000 rpm to 5000 rpm, with an acceleration ramp of 6000 rpm s⁻². Thin solid films so prepared show several clearly identifiable excitonic peaks and were scattering free. Their optical constants, used in the calculations herein performed, were obtained as in Ref. 46 and are available at Digital CSIC repository https://doi.org/10.20350/digitalCSIC/16079. Optical microscopy and atomic force microscopy were performed on the QD films prepared on the DBR mirror.

## Optical cavity and Gaussian defect preparation
We use a tunable microcavity configuration, consisting of a resonator comprising two separate halves. For the "top" cavity half, wet etching with concentrated HF is used to create a ~30 μm tall and ~200 μm wide mesa structure in the center of a glass substrate (1 cm × 1 cm). The mesa reduces the effective surface area of the two approaching cavity halves, therefore minimizing blocking from particle contamination inside the tunable resonator, allowing the two parts to approach on a hundred nm scale. On top of the mesa's surface, we used focused ion-beam (FIB) milling to pattern the Gaussian deformation. By means of magnetron-sputtering, 6.5 pairs of alternating dielectric quarter-wave layers of $Ta_2O_5/SiO_2$ have been deposited to fabricate a distributed Bragg reflector (DBR) that retains the morphology of the underlying substrate/pattern. The "bottom" cavity half is fabricated by spin coating the $CsPbBr_3$ QD perovskite film on a flat substrate with another DBR mirror comprising 9.5 pairs of quarter-wave layers of $Ta_2O_5/SiO_2$, using 5000 rpm for 60 s, with an acceleration of 6000 rpm s⁻² in a $N_2$ filled glovebox. Both "half cavities" are then mounted on xyz-nanopositioning stages to change the distance between them and move both halves independently with respect to the excitation beam, plus providing tilting degrees of freedom to enable parallel alignment.

## Determining the complex refractive index of CsPbBr₃ QDs with polystyrene
The complex refractive index of a thin film of the same QDs (Supplementary Fig. 10) was extracted from the fitting of the experimental reflectance and transmittance spectra of the CsPbBr3 QD layer, measured at different angles of incidence, using a Forouhi-Bloomer model of the dielectric constant[68]. The same complex refractive index has been successfully utilized previously to fit the response of metallic resonators embedding similar QD films[46].

## Optical characterization
White-light excitation, in transmission configuration, was used to conduct the strong-coupling measurement. The excitation light from a fiber-coupled halogen lamp was focused onto the sample, aligned on top of the Gaussian deformation, by a 100x microscope objective with a numerical aperture (NA) of 0.5 resulting in a beam size of around ~10 μm. We observe both the planar-cavity mode and the modes originating from the Gaussian potential in the transmission spectra due to the fact that the area of the focused white light is much larger than the area of the Gaussian deformation. Some of the emission image data below threshold was obtained using a fiber-coupled continuous-wave diode laser emitting at 405 nm. To drive the system into the condensation regime, a frequency-doubled, amplified laser at 400 nm, with 1 kHz repetition rate and approximately 150 fs pulse duration was used. The excitation pulses were coupled into a single-mode photonic-crystal fiber resulting in an almost perfect Gaussian spatial beam profile and a stretching of the pulses to several picoseconds. The excitation was again focused onto the sample by a 100x microscope objective with an NA of 0.5, resulting in a beam size of around 1.5–3 μm. For the transmission, k-space, and interferometric measurements, we collect the light exiting from the bottom cavity half with a 20x objective with an NA of 0.5. The signal is sent either to the front entrance of a monochromator coupled to a CCD for the energy-resolved measurements, or to a Michelson interferometer setup with a retroreflector in the adjustable arm path for the coherence measurements.

## Data availability

Data supporting the findings of this study are available at https://doi.org/10.5281/zenodo.15367737[69]. For any further clarification you can contact the authors.

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

## Acknowledgements

We thank the team of the IBM Binnig and Rohrer Nanotechnology Center for support with the cavity fabrication. We acknowledge funding from EU H2020 EIC Pathfinder Open project "PoLLoC" (Grant Agreement No. 899141), EU H2020 EIC Pathfinder Open project "TOPOLIGHT" (Grant Agreement No. 964770), EU H2020 MSCA-ITN project "AppQInfo" (Grant Agreement No. 956071), EU H2020 MSCA-ITN project "PERSE-PHONe" (Grant No. 956270), Swiss National Science Foundation project "Q-Light – Engineered Quantum Light Sources with Nanocrystal Assemblies" (Grant No. 200021_192308). the Spanish Ministry of Sci-ence and Innovation (Grant PID2020-116593RB-I00 and Grant PID2023-149344OB-100, funded by MCIN/AEI/10.13039/501100011033), and Junta de Andalucía (Grant P18-RT-2291 (FEDER/UE)).

## Author contributions

I.G. and D.T. performed the optical experiments, supported by D.U. and R.M. R.T. and D.N.D. synthesized the nanocrystals. C.B., L.C., D.T. and V.O. fabricated and characterized the QD films, with D.U. and R.M. performing the morphological characterization. I.G. and D.U. fabricated the nanostructured cavity. G.R., S.C.B., M.V.K., J.F.G.-L., H.M., R.F.M. and T.S. supervised the work. I.G., D.T., R.M. and D.U. analyzed the data and wrote the manuscript with contributions from all authors.

## Competing interests

The authors declare no competing interests.
