## [Transparent Peer Review file · Nature Communications]

Room-temperature cavity exciton-polariton condensation in perovskite quantum dots

Corresponding Author: Dr Thilo Stöferle

Version 0:

Reviewer comments:

Reviewer #1

(Remarks to the Author)

The authors report room-temperature polariton condensation in a tunable optical resonator with a Gaussian-shaped deformation, utilizing a colloidal CsPbBr₃ quantum dot film. They provide spectroscopic and coherence data to support their findings. However, similar polariton condensation has already been demonstrated in perovskite quantum dot films, albeit with different cavity configurations [Light: Sci. & Appl. 13, 34 (2024)]. The evidence for strong coupling and the angle-resolved PL spectra presented here are not convincing. Additionally, the polariton in their system comprises 7-10% excitonic, making it difficult to distinguish the emission from photonic lasing. Moreover, there are technical issues that must be addressed before this work can be considered for publication in Nature Communications.

(1) In Figure 1a, the authors should provide additional details about the Gaussian-shaped deformation, possibly including an SEM image, to clearly illustrate the lateral confinement structure. Additionally, they should explain how the cavity length was determined experimentally. In Figure 1b, the authors state that the air-gap length was determined by comparison with a transfer-matrix model simulation. The corresponding simulation results should be provided to support this claim.

(2) In Figure 1b, the original data is obscured by the fittings, making the dispersion difficult to discern. The authors should clarify how they identified the PC and LG00 cavity modes. Additionally, they need to explain how they distinguished the emission above the exciton energy as the upper polariton branch rather than cavity mode emission. Given that the Rabi splitting energy is only about 50 meV, and considering the damping rate of the excitons (FWHM ~80 meV) and cavity photons (quality factor ~500), the fulfillment of the strong coupling criteria is questionable.

(3) In Figure 2, the authors should present the angle-resolved spectra at higher angles and clearly label the detuning energies. They claim that the LP wavefunction consists of 7-10% excitonic components, indicating a highly non-equilibrium state with a very short polariton lifetime. In such situations, the distinction between photonic lasing in the weak coupling regime and polariton condensation in the strong coupling regime must be carefully examined.

(4) In Figure 3a, the authors need to explain why the uncondensed LG00 mode (purple) broadens as the pump fluence increases. Additionally, in Figure 3c, the linewidth of the polariton condensate emission for the LG00 mode remains unchanged across the threshold and does not broaden due to decoherence as the pump fluence further increases, which requires clarification. Given that the energy blueshift is not saturated, a noticeable broadening of the linewidth would be expected.

(5) In Figure 3d, the authors should clarify why the emission shows a significant blueshift despite the polariton having a relatively small excitonic fraction. Additionally, the blueshift versus pump fluence seems to exhibit three distinct stages, which require further explanation. Furthermore, the LG00 and LG01 modes display similar blueshift energies despite differences in polariton populations and excitonic component fractions. Why? This behavior suggests a photonic mode rather than a polariton mode, based on the presented data.

Reviewer #2

(Remarks to the Author)

The manuscript by Stöferle and coworkers describes the observation of exciton-polariton "condensation" in perovskite quantum dots. Although exciton-polariton condensation was traditionally limited to cryogenic temperatures, it has now been observed at room-temperature in several newer material systems such as nitrides, ZnO, organics and halide perovskites. This work shows that polariton condensates can even be demonstrated in thin film assemblies of perovskite nanocrystals, a nanomaterial that has been the focus of intense studies due to its excellent light-emitting characteristics and relatively high

stability. In terms of the characteristics typically desired for a polariton condensate (a low threshold or high nonlinearity), the performance is rather average. Still, this work will definitely be interesting to both specialists in the field and a broader audience. The manuscript is very well written and clear - I only have a handful of comments that do not significantly affect the conclusions of the work.

-While polariton "condensation" and "lasing" are often used interchangeably, I am somewhat uncomfortable using the word condensation for a cavity system where there is no kinetic energy (continuous degree of freedom) and only 2 relevant spatial modes. This is much closer to classical single of few-mode "laser" theory. Perhaps the terminology polariton lasing would be more appropriate.

-I suspect the results presented by the authors are correct, but the effective cavity length L used in the x-axis of Fig. 1 depends sensitively on the refractive index used for the QD layer close to resonance given that it is obtained from transfer matrix calculations showing strong coupling with this same index. It would be good to give this index in the supplementary along with details of how it was obtained. It would also be good to have the corresponding piezo voltage as a double (top) X axis to confirm the scaling is more or less linear with the piezo voltage. For example, if the fit is bad and the lengths on the right hand side are really a little bit longer than extracted, then that would place the cavity clearly in the weak coupling regime as one could draw a straight line from UP_PC to LP_PC.

-It would be good to have uncertainties on PC and especially LG00 Rabi splittings that were extracted. The uncertainty on the latter will be very high given that all of the data is just a small fraction of a linear dispersion.

-On line 193, the authors say that below threshold, the emission consists only of a single peak, but of course LG01 is also present. Perhaps they should restrict this to LG00 emission.

-The fact that the blueshift is below the coupling strength is not proof that the system is in the strong coupling regime because the authors are measuring in a highly negatively detuned regime. The shift between the "uncoupled" photon mode and the LP in that regime is probably very small. A blueshift of that order might mean that the transition to weak coupling occurs. However, the observed shift is tiny (1.5 meV) so I think that the conclusion stays valid.

-It makes no sense to talk about "macroscopic phase coherence" in a system with only 2 spatial modes as a hallmark of condensation. This is no different from spatial coherence in a laser resonator. In the polariton condensation case, the unique feature is the continuous highly multimode character.

-What leads to the nonlinear increase in emission? Is this light being emitted a very high angles through the stopband? Or is the PLQY of the film low? In the limiting case where the PLQY is high and there are only the 2 spatial modes described in the paper, the increase in emission across threshold should be very small (essentially just "gaining" the photons from LG01). The nonlinear increase is quite apparent - so where are the extra photons coming from?

In sum, this is a good paper and I think that it is at a novelty level and broad interest level fit for Nature Communications. The performance is not ground breaking, but this is still an impressive demonstration (and only a first). I have several comments above, but they all concern quite minor points.

Reviewer #3

(Remarks to the Author)

In this manuscript, the authors demonstrate polariton condensation achieved through perovskite quantum dots coupled with a tunable optical resonator at room temperature. The evidence provided includes strong coupling of quantum dots with a microcavity and the observation of condensation through characteristic signatures such as superlinear intensity increase, linewidth narrowing, energy blueshift, and extended temporal coherence.

While the results are interesting, it is worth noting that strong coupling in perovskites and lasing in perovskite quantum dots have been reported in prior studies. It would be beneficial if the authors provided a more detailed discussion comparing their findings with the existing literature. Below are some suggestions for improvement to be addressed before further consideration of this article:

1. Condensation and lasing

The article would benefit from a deeper discussion on the nature of condensation in this quantum system. Specifically, how do the presented results distinguish polariton condensation from conventional lasing?

Could the authors provide evidence confirming that the system remains in the strong coupling regime at pump powers exceeding the threshold?

2. Why does polariton condensation occur in the LG01 mode instead of the LG00 mode? A detailed discussion addressing this observation would be helpful. Additionally, could the authors estimate the polariton interaction constant in this system to provide further insight into the underlying physical mechanisms?

3. The authors present angular dispersion below and above the threshold (Figures 2a and 2b) and real-space images (Figures 2c and 2d). However, Figure 2c (below threshold) uses continuous-wave (CW) excitation, while Figure 2d (above threshold) employs pulsed excitation. Could the authors clarify the reasoning for using different excitation methods in these cases?

4. In Figure 3, the linewidth narrowing around the threshold is not clearly observable in Figure 3c. Could the authors explain

this discrepancy?

Version 1:

Reviewer comments:

Reviewer #1

(Remarks to the Author)

The authors have responded to the previous review by incorporating additional figures and discussions, and by comparing their work with two other QD condensate studies to assert its superiority. However, I remain concerned about the significance and rigorosity of the work, particularly in light of the critical data on light-matter strong coupling and the presentation and analysis of power dependence. The following points may need to be addressed before considering publication in Nature Communications.

Firstly, the authors should elaborate on the importance of QD film polariton condensate from two key aspects to support the significance of their work. The condensation reported is achieved in an ensemble of QDs rather than at the single or few-QD level, raising questions about its significance compared to numerous studies reporting room temperature polariton condensation in perovskite microcrystals. Does this QD ensemble condensate offer more robust quantum features against dephasing, enabling further exploration of such interacting quantum fluids of light, or does it remain within the classical regime of coherence? Secondly, QDs have been widely utilized as superior gain materials and QD lasers have been realized in VCSEL configurations similar to this work. What distinguishes the achievement of polariton condensation from photonic lasing? Except for the phase coherence the authors demonstrated, the intensity super-linear growth and linewidth narrowing are identical with that of the photon laser. And divergence angle narrowing feature is missing in this work considering the trapped geometry that broadens wavevector distribution of polariton condensate.

Some detailed comments are:

1. The authors have added strong coupling anti-crossing data in the revised Supplementary information. However, in Figure 2, which is the primary data of the work stating the formation of polariton condensate, the polariton branch with quantized energy states and anti-crossing at excitonic energy is not visible. An angle-resolved spectra with broader wavelength distribution (similar to the wavelength range in Extended Figure 8) would be needed to convince the formation of exciton polaritons in the Gaussian-defect cavity, rather than solely with cavity length-dependent measurement in Extended Figure 8. Such measurement is feasible as reported by other open-cavity works (Nature Communications, 12, 4933 (2021)).
2. As the primary data showing pump fluence dependent condensation process in the cavity polariton, the data quality in Figure 2 should be improved a lot. At least 3 high quality angle-resolved PL spectra with high signal-to-noise ratio would be needed to demonstrate the evolution from below threshold to above threshold with evident blueshift and linewidth narrowing with intensity counts labelled on each color bar, instead of showing just 2 quantized cavity modes at 2 pump fluences.
3. The authors claimed that the blueshift does not saturate and its magnitude stays significantly below coupling strength, proving that samples remain in strong coupling regime at high pump fluence. However, in response to last time's concern on interpretation of the large blueshift behavior while possessing small exciton fraction, the authors stated that the blueshift stems from various reasons (excitation-induced changes in the refractive index and partly phase-space filling effects of the optical transitions, interactions with the exciton reservoir), therefore one can have an appreciable blueshift in the system, even if the polariton has a small exciton fraction. It is challenging to reconcile these two claims. How can the origin of the one part of the blueshift be clearly attributed to polariton interactions, given the significant contributions from other factors? Additional experiments, such as resonant excitation reflectivity measurements (Nature 591, 61–65 (2021)), may be necessary to elucidate the origin of the blueshift. Furthermore, considering the open cavity geometry, it might be feasible to tune the cavity to obtain samples with larger excitonic fractions.

Reviewer #2

(Remarks to the Author)

I am satisfied with the changes. I still disagree with the condensation terminology for a trapping potential where only 2 modes are relevant - this is very different from the examples given by the authors such as the Science paper or cold atom traps for BEC which are still highly multimode, but this is mostly semantics and abused quite a bit by others as well. Overall I support publication.

Reviewer #3

(Remarks to the Author)

The authors have addressed all of my queries and recommendations, and the paper's structure now meets the publication standards of Nature Communications.

Version 2:

Reviewer comments:

Reviewer #1

(Remarks to the Author)

I recommend the publication of this version of manuscript without further revision.

Response to the Reviewers

We sincerely thank the Reviewers for their detailed reading of the manuscript, constructive and helpful comments, and overall positive assessment of our work. We followed their suggestions in close detail to improve the manuscript. Below, please find the revisions in response to the Reviewers' comments point-by-point, with the comments of the Reviewers highlighted in blue and italic, followed by our reply and action. Modifications of the manuscript are highlighted in yellow.

Reviewer: 1

Comments:

The authors report room-temperature polariton condensation in a tunable optical resonator with a Gaussian-shaped deformation, utilizing a colloidal CsPbBr₃ quantum dot film. They provide spectroscopic and coherence data to support their findings. However, similar polariton condensation has already been demonstrated in perovskite quantum dot films, albeit with different cavity configurations [Light: Sci. & Appl. 13, 34 (2024)]. The evidence for strong coupling and the angle-resolved PL spectra presented here are not convincing. Additionally, the polariton in their system comprises 7-10% excitonic, making it difficult to distinguish the emission from photonic lasing. Moreover, there are technical issues that must be addressed before this work can be considered for publication in Nature Communications.

Our response: We thank the reviewer for the constructive feedback.

We agree that polariton condensation in a perovskite quantum dot film has already been claimed in [Light: Sci. & Appl. 13, 34 (2024), <https://doi.org/10.1038/s41377-024-01378-5>], which we had already cited (as Ref. 41) and briefly discussed in the manuscript. However, our submitted manuscript has two fundamental differences to the aforementioned work, which we believe constitute remarkable developments that are significant steps forward and will attract broad interest and seed further experiments, studies and applications. First, our work is conducted at room-temperature instead of cryogenic temperatures ($T = 10$ K) used in [Light: Sci. & Appl. 13, 34 (2024)]. The ability of operating at ambient conditions is of utmost importance towards the realization of practical devices (was Ref. 46, now Ref. 48, <https://doi.org/10.1038/nmat4668>). Second, in our work we use an optical microcavity, thereby creating cavity exciton-polaritons, while [Light: Sci. & Appl. 13, 34 (2024)] uses a waveguide creating waveguide exciton-polaritons. This difference is not subtle and leads to a fundamental distinction in the condensation process between the two systems: In [Light: Sci. & Appl. 13, 34 (2024)], the condensation in a waveguide geometry does not occur in the ground state of the system but in an excited, propagating mode ($k \gg 0$), while in our microcavity structure the polariton condensation occurs in the ground state near zero quasi-momentum ($k = 0$), which is consistent with traditional systems exhibiting Bose-Einstein condensation such as cold atoms, and moreover, is the geometry that allows compact polaritonic devices. For these reasons, we are confident that our work poses a significant advancement in the fields of ambient condition polaritonics and perovskite quantum dots. Furthermore, we trust that with the additional data and analysis presented in the following, the concerns about the strong coupling regime and polariton condensation are sufficiently addressed.

Our action: To further highlight the fundamental differences of our work and previous work claiming condensation [Light: Sci. & Appl. 13, 34 (2024)] we have added the following sentence to the manuscript.

“In contrast to the previously reported observation where waveguide polaritons condense at $T = 10$ K in an excited, propagating state⁴¹, in our work cavity polaritons condense at room temperature in the ground state of the polariton branch, consistent with condensation processes in traditional bosonic platforms.”

(1) In Figure 1a, the authors should provide additional details about the Gaussian-shaped deformation, possibly including an SEM image, to clearly illustrate the lateral confinement structure. Additionally, they should explain how the cavity length was determined experimentally. In Figure 1b, the authors state that the air-gap length was determined by comparison with a transfer-matrix model simulation. The corresponding simulation results should be provided to support this claim.

Our response: We thank the Reviewer for this suggestion. However, the reviewer might have overlooked the AFM image of the Gaussian-shaped deformation displayed in Extended Data Fig. 4. We believe that this provides much more useful information than an SEM image could convey, as it allows assessment of the surface shape and roughness with nanometer resolution. Furthermore, the transfer matrix simulation results are shown in Extended Data Fig. 6 (now modified and changed to Extended Data Fig. 8). The caption of this Extended Data Figure explains how the cavity length was determined utilizing the transfer matrix simulation results. Thanks to the reviewer’s comment and after revisiting the analysis of the simulation results, we decided to slightly adjust the determined cavity length. That is due to the fact that the attribution of the experimentally extracted peak positions to a certain longitudinal order is not completely unambiguous. This adjustment that corresponds to a change in the longitudinal mode order number does not affect in any way the conclusions related to the underlying physics in this work.

Our action: We have modified the Extended Data Fig. 6 (now Extended Data Fig. 8) to explain the calibration procedure better and adjusted the determined cavity length throughout the manuscript.

Extended Data Figure 8 | Transfer matrix simulation for cavity length calibration. **a** The left panel is the same measurement present in Fig. 1b but without the fits. The right panel is again the same measurement but additionally the extracted peak positions are highlighted with black circles for the PC mode and white circles for the LG00 mode. The top axis shows the raw closed-loop z-positioner value, which is translated through a lever-like motion into a much smaller change of cavity length. **b**, Transfer-matrix simulation of transmission spectra for a wide range of cavity lengths which includes the 2nd, 3rd and 4th longitudinal order cavity modes. By comparing the extracted data from the experimental polariton dispersion of the PC mode shown in (a) (black circles) to the simulation, we conclude that the experimental data match slightly better the region highlighted with the green dashed box compared to the red dashed box regions, therefore determining the length L of our cavity. **c**, Zoomed-in images of the areas inside the red and green dashed boxes from panel (b). Here, it becomes even more apparent that the overlap between the extracted experimental data and the simulated dispersion inside the green box area pair the best. **d**, Extracted experimental data from the transmission measurement, showing both the 3rd and 4th

longitudinal order PC modes. The 4th order mode, which is far from the exciton, tunes linearly which indicates linear pressing between the cavity halves and true linear, lever-like modification of the cavity length.

(2) In Figure 1b, the original data is obscured by the fittings, making the dispersion difficult to discern. The authors should clarify how they identified the PC and LG00 cavity modes. Additionally, they need to explain how they distinguished the emission above the exciton energy as the upper polariton branch rather than cavity mode emission. Given that the Rabi splitting energy is only about 50 meV, and considering the damping rate of the excitons (FWHM ~80 meV) and cavity photons (quality factor ~500), the fulfillment of the strong coupling criteria is questionable.

Our response: We agree with the remarks of the Reviewer regarding the visibility of the data in Fig. 1b and about the distinction between the PC and LG00 cavity modes. Therefore, we have slightly revised Fig. 1b of the manuscript, increasing the transparency of the fitted lines. We have also added a new Extended Data Fig. 6 which shows two additional transmission measurements demonstrating strong coupling, one in a different position of the sample and a second one measured on a second sample, to further confirm our claim of the system being in the strong coupling regime. Additionally, we have added Extended Data Fig. 7 that shows the characteristic k-space emission of multiple states (obtained at larger cavity detuning), allowing to identify PC and LG modes. Below, the left panel of Extended Data Fig. 8a shows the strong coupling measurement displayed in Fig. 1b but now with the fittings removed. There it becomes quite clear that the upper and the lower branch cannot be seamlessly connected, as is expected from an anti-crossing behavior. This is also seen in the waterfall plot in Extended Data Fig. 5. The only way to determine that the state energetically above the exciton really corresponds to the upper polariton branch is to fit the dispersion with the coupled oscillator model. Due to the good agreement of our performed fitting (Fig. 1b and Extended Data Fig. 5) we are confident that the emission can be attributed to the upper polariton.

The Reviewer makes a solid point comparing the determined Rabi splitting and the decay rates of the excitons and cavity photons. Nevertheless, it is important to distinguish the broadened emission linewidth of the ensemble exciton emission (FWHM ~80 meV) with the actual decay rate of the exciton. In CsPbBr₃ quantum dots, the typical lifetime is around 3 - 5 ns at room temperature (see new Ref. 42, <https://doi.org/10.1021/acsnano.2c07677> and new Ref. 43, <https://doi.org/10.1021/acscentsci.0c01153>) corresponding to a 0.7 μeV decay rate. As already mentioned in the manuscript, the cavity quality factor of ~500 corresponds to a ~5 meV decay rate, which renders both exciton and cavity photon decay rates lower than the observed Rabi splitting. However, inhomogeneous broadening can reduce or even obscure an anti-crossing while the system is still in the strong coupling regime, see (was Ref. 45, now Ref. 47. <https://doi.org/10.1515/nanoph-2024-0049>).

For these reasons we believe that our strong coupling claim is sufficiently robust but at the same time we have revised the manuscript to address the Reviewer's concerns.

Extended Data Figure 8a (left panel) | Strong coupling measurement without fits.

Our action: We have changed panel **b** of Figure 1 with a revised version where the fitted dispersions are now slightly transparent to allow for better visualization of the underlying, measured data.

Figure 1

We have added Extended Data Fig. 6 that presents two additional transmission measurements demonstrating strong coupling.

Extended Data Figure 6 | Additional white light transmission measurements demonstrating strong coupling. **a**, Transmission measurement taken at a different position of the main sample as the one used in the manuscript, where we see the anticrossing of the PC mode with a fitted Rabi splitting of (87 ± 2) meV. **b**, Transmission measurement on a second, similar sample utilizing the same CsPbBr₃ QDs. Here, we can observe the level tuning for both PC and LG modes. The fitted Rabi splitting for the PC mode is (59 ± 2) meV. In both panels the fits to the polariton dispersion are represented by the red solid lines, while the exciton and photon energies are given by the dashed orange lines. The extracted positions of the several peaks are indicated with the black circles. The raw closed-loop z-stage positions are shown as horizontal axes, and the marked difference between the samples is a result of different pressing / lever arm conditions when the top and bottom substrates are in contact.

We have added Extended Data Fig. 7 to show how we identify the LG00 and PC modes in our measurements based on their k -space characteristics.

Extended Data Figure 7 | Cavity dispersion for identifying the various photonic modes. The angular dispersion measured at large detuning from the exciton reveals the set of Laguerre-Gaussian modes originating from the Gaussian-shaped deformation and the planar cavity mode. As shown in the image, the lowest energy dispersionless mode corresponds to the LG00 mode

while the highest energy parabolic mode corresponds to the planar cavity mode PC. On the left side of the panel, the respective angle-integrated spectrum (black solid line) is presented.

To clarify the discussion about the excitonic transition damping rate we have removed the parenthesis “(width is dominated by homogeneous broadening at room temperature)”, and we have added the following sentence:

The exciton spectral width is dominated by a combination of inhomogeneous and homogeneous broadening at room temperature³⁹, while the actual damping rate of the transition is around ~ 1 μeV corresponding to the reported 3 – 5 ns lifetime of CsPbBr₃ QDs^{22,42,43}.

(3) In Figure 2, the authors should present the angle-resolved spectra at higher angles and clearly label the detuning energies. They claim that the LP wavefunction consists of 7-10% excitonic components, indicating a highly non-equilibrium state with a very short polariton lifetime. In such situations, the distinction between photonic lasing in the weak coupling regime and polariton condensation in the strong coupling regime must be carefully examined

Our response: The displayed angle range in Figure 2 contains all relevant experimental signals, which is also evident from the new Extended Data Fig. 7 presented above, where one can additionally see the planar cavity dispersion. The discussion related to the question “is it a laser or a condensate” has been an ongoing debate in the community for the last decades (see Ref.: Nature 447, 540 (2007) (<https://doi.org/10.1038/447540a>), Nature Photonics 6, 2 (2012) (<https://doi.org/10.1038/nphoton.2011.325>), Nature Photonics 6, 205 (2012) (<https://doi.org/10.1038/nphoton.2012.52>), Nature Physics 10, 803 (2014) (<https://doi.org/10.1038/nphys3143>)) where terms such as “photon laser, photon BEC, polariton laser, polariton BEC” have been often used interchangeably to describe similar effects. Here, we can clearly see that even though the LG00 (2.38 eV / 2.39 eV in Fig. 2 / 3) should be within the optical gain range because it is still only in the tail of the QD absorption spectrum, the polariton condensation only occurs in the lowest polariton branch (LG00), even though a higher energy state (LG01 at 2.40 eV / 2.41 eV in Fig. 2 / 3) is closer to the PL peak emission (Fig. 1b, right panel). One would anticipate (multimode-) lasing in such energetically higher states from a photon laser, at least at greater excitation fluence. Furthermore, even at the highest pump fluence, the polariton blueshift as indicated in Figure 3c (~ 4 meV) remains significantly below the energy difference between the LG00 lower polariton and photonic mode (~ 8 meV for the data in Figure 2 and ~ 10 meV for the data in Figure 3). In addition, the blueshift does not saturate, which suggests that the system remains in the strong coupling regime even far above threshold. Furthermore, as already mentioned in the manuscript, we believe that the observation that the blueshift of the emission starts near or just below the condensation threshold is another evidence pointing towards polariton condensation.

Our action: For clarifying the Reviewer’s remark regarding the labeling of the detuning energies we changed the sentence “... corresponding to a detuning of the LG00 cavity mode from the exciton of -100 meV.” with the highlighted sentence below.

“... corresponding to LG00 exciton-photon detuning of -100 meV and a detuning of -72 meV for the PC mode.”

Additionally, we have modified the caption of Figure 3, explicitly mentioning the detunings for the PC and LG00 modes by adding the following sentence.

“At condensation threshold the PC and LG00 exciton-photon detunings are -57 meV and -85 meV, respectively.”

On top of that, to further address that the system remains in the strong coupling regime above threshold we have modified the sentence “Nevertheless, the magnitude of the blueshift stays significantly below the coupling strength and therefore proves that, even for the highest excitation fluence, the sample remains in the strong light–matter coupling regime.” to the one below:

“Nevertheless, the blueshift does not saturate and its magnitude stays significantly below both the coupling strength (60 meV) and the energy difference between the LG00 lower polariton and photonic mode (~10 meV), therefore proving that, even for the highest studied excitation fluence, the sample remains in the strong light–matter coupling regime.”

(4) In Figure 3a, the authors need to explain why the uncondensed LG00 mode (purple) broadens as the pump fluence increases. Additionally, in Figure 3c, the linewidth of the polariton condensate emission for the LG00 mode remains unchanged across the threshold and does not broaden due to decoherence as the pump fluence further increases, which requires clarification. Given that the energy blueshift is not saturated, a noticeable broadening of the linewidth would be expected.

Our response: As the Reviewer points out, in Figure 3a it looks like for higher pump fluence the uncondensed LG00 mode (purple) broadens. Yet, as one can see from Figure 3c the broadening of the mode near and above threshold is within the error bars from the fit error, which does not allow to draw conclusions. We believe that the broadening of the condensate’s linewidth for increasing pump fluence that the reviewer anticipates should be expected for condensation in a planar microcavity but it might be different in a trapping potential like ours. Due to the parabolic cavity dispersion in a planar microcavity, there is a continuum of states available, allowing interacting polaritons to easily occupy broader distributions in quasi-momentum and energy, resulting in a broadening of the linewidth. In contrast, the Gaussian-shaped deformation provides a limited set of discrete energy states, meaning that there are no very nearby states (within few meV) for the polaritons to populate (the energy difference between LG00 and LG01 is around 20 meV). Hence, the emission width should be ideally limited by the polariton coherence but in reality will be influenced also by noise. While this provides a rationale for the observed behavior, we believe – in-line with the Reviewer question – that in general polariton condensation in lower-dimensional systems with strong discretization in both or just one lateral dimension still holds many interesting questions that could inspire future studies, with QDs or other materials.

(5) In Figure 3d, the authors should clarify why the emission shows a significant blueshift despite the polariton having a relatively small excitonic fraction. Additionally, the blueshift versus pump fluence seems to exhibit three distinct stages, which require further explanation. Furthermore, the LG00 and LG01 modes display similar blueshift energies despite differences in polariton populations and excitonic component fractions. Why? This behavior suggests a photonic mode rather than a polariton mode, based on the presented data.

Our response: We thank the reviewer for this comment. As we mention in the manuscript, we attribute the condensate’s blueshift to a variety of reasons (excitation-induced changes in the

refractive index and partly phase-space filling effects of the optical transitions, interactions with the exciton reservoir) where one of them is also polariton-polariton interactions. This means that one can have an appreciable blueshift in the system, even if the polariton has a small exciton fraction (see old Ref. 47, now Ref. 49, ACS Photonics (2018) (<https://doi.org/10.1021/acsp Photonics.7b00557>) and old Ref. 49, now Ref. 51 Commun Phys (2020) (<https://doi.org/10.1038/s42005-019-0278-6>)). As we argue in the manuscript, the origin of the blueshift in this system is still unclear and can also vary at different spatial positions of the sample due to material inhomogeneity and variation of the cavity length. We believe that what looks like distinct stages of the blueshift versus the pump fluence in Figure 3d could be artifacts from slight drift or instability of the open cavity configuration, because – while the general observations like threshold-like behavior, narrowing and blueshift are reproduced on other sample positions – the detailed behaviors are varying a bit between different positions and samples (see new Extended Data Fig. 9). Therefore, we would not overinterpret the curve shape of the blueshift. However, it is important to note that we have consistently observed a blueshift of the condensate peak emission.

Our action: We have added the Extended Data Fig. 9, which shows one additional threshold measurement at a different position of the main sample (Extended Data Fig. 9a) and two measurements at different positions from an almost identical sample to the main one (Extended Data Fig. 9b,c) to showcase that while there can be some variations, the main effects (threshold, narrowing, blueshift) are robust and remain quite similar. The figure is presented below.

Extended Data Figure 9 | Additional condensation threshold measurements. *a*, Additional threshold measurement at a different position of the main sample. *b-c*, Two extra measurements at different spots on a second, almost identical sample. The uncondensed LG00 polariton emission is denoted by purple discs, the LG00 condensate by orange squares, and the non-condensing LG01 emission by grey diamonds. The panels within in each show (top) emission intensity, (middle) emission linewidth, (bottom) emission peak energy. Error bars obtained from the fit errors are only significant for the uncondensed LG00 polariton emission linewidth. All three measurements show nonlinear increase of the emission and linewidth narrowing at threshold, and a blueshift of the condensate's emission. As mentioned in the main text, the threshold, linewidth and blueshift

behavior can change between different spots, due to either material inhomogeneities and/or cavity variations (Q factor, longitudinal mode order etc.).

Additionally, we have edited the following sentences in the manuscript.

“It is important to note that while the main effects remain quite similar, repeating the excitation-dependent measurements at different material positions and at a second, almost identical, sample led to variations of mode energy, threshold behavior and blueshift, as a result of both film inhomogeneity and small fluctuations of the geometry of the experimental setup (e.g., drift of the cavity length or position) (Extended Data Fig. 9).”

Reviewer: 2

Comments:

The manuscript by Stoferle and coworkers describes the observation of exciton-polariton "condensation" in perovskite quantum dots. Although exciton-polariton condensation was traditionally limited to cryogenic temperatures, it has now been observed at room-temperature in several newer material systems such as nitrides, ZnO, organics and halide perovskites. This work shows that polariton condensates can even be demonstrated in thin film assemblies of perovskite nanocrystals, a nanomaterial that has been the focus of intense studies due to its excellent light-emitting characteristics and relatively high stability. In terms of the characteristics typically desired for a polariton condensate (a low threshold or high nonlinearity), the performance is rather average. Still, this work will definitely be interesting to both specialists in the field and a broader audience. The manuscript is very well written and clear - I only have a handful of comments that do not significantly affect the conclusions of the work.

In sum, this is a good paper and I think that it is at a novelty level and broad interest level fit for Nature Communications. The performance is not ground breaking, but this is still an impressive demonstration (and only a first). I have several comments above, but they all concern quite minor points.

Our response: We thank the Reviewer for the positive assessment and acknowledging the novelty and impact of our work. We concur that in terms of current threshold/nonlinearity metrics this first demonstration of a cavity exciton-polariton condensate with quantum dots is indeed not ground breaking, but in-line with the Reviewer, we strongly believe that the results will be of broad, interdisciplinary interest and set a solid basis for future work to further optimize the colloidal nanocrystal material and cavities.

-While polariton "condensation" and "lasing" are often used interchangeably, I am somewhat uncomfortable using the word condensation for a cavity system where there is no kinetic energy (continuous degree of freedom) and only 2 relevant spatial modes. This is much closer to classical single of few-mode "laser" theory. Perhaps the terminology polariton lasing would be more appropriate.

Our response: While we understand the concern of the Reviewer we would like to point out two main reasons why we believe the term “polariton condensation” is justified here. First, as the

Reviewer mentioned already, the two terms have been used interchangeably in literature and the term “polariton condensation” has already been widely used by the polariton community in similar systems where a trapping potential for the polariton wavefunction is implemented (see Ref.: Science (2007) (<https://www.science.org/doi/10.1126/science.1140990>), Nature Physics (2013) (<https://doi.org/10.1038/nphys2609>), Physical Review Letters (2012) (<https://doi.org/10.1103/PhysRevLett.108.126403>), Nature Physics (2024) (<https://doi.org/10.1038/s41567-023-02281-3>), was Ref. 47, now Ref. 49, (<https://doi.org/10.1021/acsphotonics.7b00557>)). Second, in traditional bosonic systems exhibiting Bose-Einstein condensation such as cold atoms, the atoms are also confined in traps with discrete trap levels, and actually the finite-size effects arguably allow even much richer physics than in the thermodynamic limit (see, e.g. Rev. Mod. Phys. (1999) (<https://doi.org/10.1103/RevModPhys.71.463>)).

-I suspect the results presented by the authors are correct, but the effective cavity length L used in the x -axis of Fig. 1 depends sensitively on the refractive index used for the QD layer close to resonance given that it is obtained from transfer matrix calculations showing strong coupling with this same index. It would be good to give this index in the supplementary along with details of how it was obtained. It would also be good to have the corresponding piezo voltage as a double (top) X axis to confirm the scaling is more or less linear with the piezo voltage. For example, if the fit is bad and the lengths on the right hand side are really a little bit longer than extracted, then that would place the cavity clearly in the weak coupling regime as one could draw a straight line from UP_PC to LP_PC.

Our response: We thank the Reviewer for the suggestions. We have added a section in the Methods and an extra figure as Extended Data Fig. 10 regarding the used refractive index. The positioner stage that we utilize for the cavity tuning is operating in closed-loop (Smaract SLC-1720 with 1 nm sensor resolution, operated in hybrid stick-slip/scan mode). Instead of voltage, we can monitor the relative position value of the stage. Notably, as both cavity halves are in contact (not directly at the position of the Gaussian but at a lateral distance), moving the z -stage by micrometers results in a lever-like translation to much smaller changes in cavity length on the order of (tens of) nanometers. To check that this translated cavity length tuning is linear with the stage encoder position, we monitor the tuning of a cavity mode that is far away of the exciton and therefore is expected to tune in a linear way across the whole measurement. We have added panel (d) in the new Extended Data Fig. 8 showcasing this point and explaining it in the figure’s caption, together with the whole cavity length calibration procedure.

Our action: Regarding the refractive index comment we have added the following part in the Methods section.

Determining the complex refractive index of CsPbBr₃ QDs with polystyrene. The complex refractive index of a thin film of the same QDs (Extended Data Fig. 10) was extracted from the fitting of the experimental reflectance and transmittance spectra of the CsPbBr₃ QD layer, measured at different angles of incidence, using a Forouhi-Bloomer model of the dielectric constant⁵⁹. The same complex refractive index has been successfully utilized previously to fit the response of metallic resonators embedding similar QD films⁴⁰.

and the following figure as Extended Data Fig. 10.

Extended Data Figure 10 | Real (n) and imaginary (κ) parts of the refractive index of a film of CsPbBr₃ QDs with polystyrene, in blue and red respectively. The presented measured components of the complex refractive index were used in the transfer matrix simulation presented in Extended Data Figure 8.

To address the scaling of the horizontal axis for the strong coupling measurement, we have added the following panel (d) in the new Extended Data Fig. 8.

Extended Data Figure 8d | Extracted experimental data from the transmission measurement, showing both the 3rd and 4th longitudinal order PC modes. The 4th order mode, which is far from the exciton, tunes linearly which indicates linear pressing and modification of the cavity length.

-It would be good to have uncertainties on PC and especially LG00 Rabi splittings that were extracted. The uncertainty on the latter will be very high given that all of the data is just a small fraction of a linear dispersion.

Our response: We appreciate the suggestion and agree with the Reviewer's comment. From our fits we extracted the values (53 ± 2) meV and (60 ± 2) meV for the PC and LG00 modes, respectively,

where the error indicate the fit error. The reason that the LG00 mode has a similar error as the PC mode is that we fitted two coupled oscillator models for PC and LG00 modes simultaneously, while having the same exciton energy and the same slope of the photon mode (vs. cavity length) for both. This resulted in quite accurate estimations of coupling strengths and the exciton and uncoupled photon energies, mainly because of the presence of both lower and upper polariton branch for the PC mode.

Our action: We have added the errors to the presented values in the main manuscript.

-On line 193, the authors say that below threshold, the emission consists only of a single peak, but of course LG01 is also present. Perhaps they should restrict this to LG00 emission.

Our response: We thank the reviewer for the remark, which we fully agree with, and therefore, we have revised the manuscript accordingly.

Our action: We have modified the respective sentences to the ones below.

“Below condensation threshold, the **LG00 emission** spectrum consists of a single peak, attributed to uncondensed polaritons.”

and

“Notably, for below-threshold and above-threshold **LG00 emission** spectra exhibiting a single resolvable peak only, we fitted the spectrum with a single Gaussian function, while near the threshold we performed a fit consisting of the sum of two Gaussians.”

-The fact that the blueshift is below the coupling strength is not proof that the system is in the strong coupling regime because the authors are measuring in a highly negatively detuned regime. The shift between the "uncoupled" photon mode and the LP in that regime is probably very small. A blueshift of that order might mean that the transition to weak coupling occurs. However, the observed shift is tiny (1.5 meV) so I think that the conclusion stays valid.

Our response: We agree with the Reviewer’s comment that here a comparison with the uncoupled photon mode would be appropriate. We have revised the manuscript, so it becomes clear that even far above threshold the blueshift (~4 meV) remains both below the coupling strength (60 meV) and the difference between the "uncoupled" photon mode and the polariton mode (10 meV).

Our action: We have included this information in the respective sentence.

“Nevertheless, the blueshift does not saturate and its magnitude stays significantly below both the coupling strength (60 meV) and the energy difference between the LG00 lower polariton and photonic mode (~10 meV), therefore proving that, even for the highest studied excitation fluence, the sample remains in the strong light–matter coupling regime.”

-It makes no sense to talk about "macroscopic phase coherence" in a system with only 2 spatial modes as a hallmark of condensation. This is no different from spatial coherence in a laser resonator. In the polariton condensation case, the unique feature is the continuous highly multimode character.

Our response: Indeed, compared to our former experiments with organics (Fig. 4 in old Ref. 47, now Ref. 49, <https://pubs.acs.org/doi/abs/10.1021/acsp Photonics.7b00557>), here the change in spatial mode occupation across the threshold is less pronounced (Fig. 4a left panels), mainly due to the low signal-to-noise ratio below threshold. Hence, we have slightly modified the respective sentences.

Our action: We have modified the wording “macroscopic phase coherence” to the one below, that further relates to the temporal coherence:

“**extended** phase coherence”

Moreover, we have modified the sentence “The resulting continuous fringe pattern with constant contrast over the whole emission interferogram (see Fig. 4a) suggests that a single, macroscopic polariton condensate mode is populated with the LG00 cavity mode.” to the one below.

“The resulting continuous fringe pattern with constant contrast over the whole emission interferogram (see Fig. 4a) suggests that a single-mode polariton condensate is formed in the LG00 state.”

-What leads to the nonlinear increase in emission? Is this light being emitted at very high angles through the stopband? Or is the PLQY of the film low? In the limiting case where the PLQY is high and there are only the 2 spatial modes described in the paper, the increase in emission across threshold should be very small (essentially just "gaining" the photons from LG01). The nonlinear increase is quite apparent - so where are the extra photons coming from?

Our response: The Reviewer is perfectly right with the two mechanisms leading to the apparent nonlinear increase of the emission: It is true that there are many photons emitted from the exciton reservoir or at higher angles in the polariton dispersion that are not captured by the microscope objective in these measurements. It can sometimes be seen in the experiment that the edge of the mesa structures, where the DBR mirror ends, “glows” while the actual excitation spot is many tens of microns away on the Gaussian deformations. The fast stimulated scattering of excitations into the polariton condensate redistributes their decay channels towards the condensate mode and furthermore significantly reduces their effective decay time (from ns to ps), eventually reducing detrimental trapping and quenching mechanisms in the material. Although the quantum yield at room-temperature in these QDs can reach above 80% in solution (see Ref. 39 <https://doi.org/10.1126/science.abq3616>), it can be substantially reduced in dense, dried thin films.

Reviewer: 3

Comments:

In this manuscript, the authors demonstrate polariton condensation achieved through perovskite quantum dots coupled with a tunable optical resonator at room temperature. The evidence provided includes strong coupling of quantum dots with a microcavity and the observation of condensation through characteristic

signatures such as superlinear intensity increase, linewidth narrowing, energy blueshift, and extended temporal coherence.

While the results are interesting, it is worth noting that strong coupling in perovskites and lasing in perovskite quantum dots have been reported in prior studies. It would be beneficial if the authors provided a more detailed discussion comparing their findings with the existing literature. Below are some suggestions for improvement to be addressed before further consideration of this article:

Our response: We thank the Reviewer for acknowledging the value of our work and the constructive, detailed comments to improve the manuscript.

We agree that polariton condensation in a perovskite quantum dot film has already been claimed in [Light: Sci. & Appl. 13, 34 (2024), <https://doi.org/10.1038/s41377-024-01378-5>], which we had already cited (as Ref. 41) and briefly discussed in the manuscript. However, our submitted manuscript has two fundamental differences to the aforementioned work, which we believe constitute remarkable developments that are significant steps forward and will attract broad interest and seed further experiments, studies and applications. First, our work is conducted at room-temperature instead of cryogenic temperatures ($T = 10$ K) used in [Light: Sci. & Appl. 13, 34 (2024)]. The ability of operating at ambient conditions is an important condition towards the realization of practical devices (was Ref. 46, now Ref. 48, <https://doi.org/10.1038/nmat4668>). Second, in our work we use an optical microcavity and create cavity exciton-polaritons, while [Light: Sci. & Appl. 13, 34 (2024)] uses a waveguide and waveguide exciton-polaritons are observed. This difference is not subtle and leads to a fundamental distinction in the condensation process between the two systems: In [Light: Sci. & Appl. 13, 34 (2024)], the condensation in a waveguide structure does not occur in the ground state of the system but in an excited, propagating mode ($k \gg 0$), while in our microcavity structure the polariton condensation occurs in the ground state near zero quasi-momentum ($k = 0$), which is consistent with traditional systems exhibiting Bose-Einstein condensation such as cold atoms, and moreover, is the geometry that allows compact polaritonic devices. For these reasons, we believe that our work poses a significant advancement in the fields of ambient condition polaritonics and perovskite quantum dots.

Furthermore, one has to also consider the versatility and advantages provided from the patterned, open cavity configuration that we use. The open nature of the cavity allows for a high degree of tunability and high cavity finesse and therefore can be crucial going towards the few/single quantum dot regime. Finally, demonstrating condensation in a patterned potential opens the door for using perovskite quantum dots together with arbitrary engineered potential landscapes towards room temperature analogue simulations, as showcased with organic materials, e.g., in <https://doi.org/10.1038/s42005-021-00548-w>.

Our action: In order to further highlight the fundamental differences of our work and previous work claiming condensation [Light: Sci. & Appl. 13, 34 (2024)] we have added the following sentence in the manuscript.

“In contrast to the previously reported observation where waveguide polaritons condense at $T = 10$ K in an excited, propagating state⁴¹, in our work cavity polaritons condense at room temperature in the ground state of the polariton branch, consistent with condensation processes in traditional bosonic platforms.”

1. Condensation and lasing

☐ *The article would benefit from a deeper discussion on the nature of condensation in this quantum system. Specifically, how do the presented results distinguish polariton condensation from conventional lasing?*

☐ *Could the authors provide evidence confirming that the system remains in the strong coupling regime at pump powers exceeding the threshold?*

Our response: We appreciate the Reviewer's comments. The discussion related to the question "is it a laser or a condensate" has been an ongoing debate in the community for the last decades (see Ref.: Nature 447, 540 (2007) (<https://doi.org/10.1038/447540a>), Nature Photonics 6, 2 (2012) (<https://doi.org/10.1038/nphoton.2011.325>), Nature Photonics 6, 205 (2012) (<https://doi.org/10.1038/nphoton.2012.52>), Nature Physics 10, 803 (2014) (<https://doi.org/10.1038/nphys3143>)) where terms such as "photon laser, photon BEC, polariton laser, polariton BEC" have been often used interchangeably to describe similar effects. Here, we can clearly see that even though the LG00 (2.38 eV / 2.39 eV in Fig. 2 / 3) should be within the optical gain range because it is still only in the tail of the QD absorption spectrum, the polariton condensation only occurs in the lowest polariton branch (LG00), even though a higher energy state (LG01 at 2.40 eV / 2.41 eV in Fig. 2 / 3) is closer to the PL peak emission (Fig. 1b, right panel). From a photon laser one would expect (multimode-)lasing in such energetically higher states, at least at higher excitation fluence. Furthermore, even at the highest pump fluence, the polariton blueshift as indicated in Figure 3c (~4 meV) remains significantly below the energy difference between the LG00 lower polariton and photonic mode (~8 meV for the data in Figure 2 and ~10 meV for the data in Figure 3), and the blueshift does not saturate, which suggests that the system remains in the strong coupling regime even far above threshold. Additionally, as already mentioned in the manuscript, we believe that the observation that the blueshift of the emission starts near or just below the condensation threshold can be another evidence pointing towards polariton condensation.

Our action: To clarify that the system remains in the strong coupling regime above threshold we have modified the sentence "Nevertheless, the magnitude of the blueshift stays significantly below the coupling strength and therefore proves that, even for the highest excitation fluence, the sample remains in the strong light-matter coupling regime." to the one below:

"Nevertheless, the blueshift does not saturate and its magnitude stays significantly below both the coupling strength (60 meV) and the energy difference between the LG00 lower polariton and photonic mode (~10 meV), therefore proving that, even for the highest studied excitation fluence, the sample remains in the strong light-matter coupling regime."

2. *Why does polariton condensation occur in the LG01 mode instead of the LG00 mode? A detailed discussion addressing this observation would be helpful. Additionally, could the authors estimate the polariton interaction constant in this system to provide further insight into the underlying physical mechanisms?*

Our response: There is a chance that the Reviewer has overlooked that polariton condensation occurs in the LG00 state and not in the LG01 state. This is therefore consistent with the condensation process where bosons form a condensate in the energetically lowest state.

The question about the polariton interactions is very interesting, however, we unfortunately cannot give a solid answer based on the current experiments in this open-cavity system. As

mentioned in the manuscript, with the present geometry, the observed blueshift can be attributed to several mechanisms like excitation-induced changes in the refractive index and saturation effects of the optical transitions, interactions with the exciton reservoir and also polariton-polariton interactions. To decouple the polariton-polariton interactions from the other effects would require specific experimental methods that have been explored with inorganic semiconductor polariton condensates, such as spatially separating the exciton reservoir from the polariton condensate. But these measurements required excellent quantum wells and high-Q cavities within much larger, confined structures (see Phys. Rev. B (2011) (<https://doi.org/10.1103/PhysRevB.84.195301>), Phys. Rev. Lett. (2011) (<https://doi.org/10.1103/PhysRevLett.106.126401>), Nature Photonics (2022) (<https://doi.org/10.1038/s41566-022-01019-6>)). With the comparatively rough perovskite QD films, this certainly would constitute another project by itself in order to obtain reliable data on the polariton interaction constant.

3. The authors present angular dispersion below and above the threshold (Figures 2a and 2b) and real-space images (Figures 2c and 2d). However, Figure 2c (below threshold) uses continuous-wave (CW) excitation, while Figure 2d (above threshold) employs pulsed excitation. Could the authors clarify the reasoning for using different excitation methods in these cases?

Our response: Here the Reviewer poses a perfectly valid question regarding the use of a different excitation method for Figure 2c. The reason is that if we had used the pulsed excitation below threshold for the real space imaging, we would have needed to integrate for a very long time in order to have a good image that allows to see the spatial structure much better than e.g. for the coherence measurement (Fig. 4a top right panel, used 10 s integration with pulsed excitation). However, as we are working in contact-mode between the two cavity halves our open-cavity setup can be slightly unstable over longer durations, especially with these comparably soft perovskite QD solids. Hence, we decided to use continuous-wave (CW) excitation at the same off-resonant excitation wavelength that allows to obtain much stronger emission while still staying below threshold density, because the pump photons are spread out much more in time than with the ultrafast, pulsed excitation. Therefore, this change in excitation source had only practical, signal-to-noise reasons, and does not involve any change in the polariton states (far below threshold).

4. In Figure 3, the linewidth narrowing around the threshold is not clearly observable in Figure 3c. Could the authors explain this discrepancy?

Our response: We thank the Reviewer for the comment. At condensation threshold, the linewidth narrowing is observed when comparing the LG00 uncondensed polaritons (purple circles) with the forming LG00 condensate. In the experiment in the excitation fluence range near the threshold, we observe both peaks, presumably due to the fact that the spectra are time-integrated, and the pulsed excitation has a considerable pulse time before the actual threshold fluence is reached, and then when the (blueshifted) condensate has decayed, still uncondensed polaritons remain. Due to the time-integrated measurement all these three “phases” are captured during the pulsed excitation in one spectrum. The linewidth corresponding to the uncondensed LG00 mode is around 1 meV, while at threshold, the polariton condensate is formed with a linewidth around 0.4 meV, as shown in Figure 3c. Hence, there is an actual drop in linewidth from 1.0 to 0.4 meV at the threshold between the uncondensed and condensed LG00 emission, respectively. Here, we have

used different colors to distinguish the condensate (orange and only above threshold) and the uncondensed polaritons (purple, below and above threshold), but in the end both colors refer to the same cavity mode (LG00).

Response to the Reviewers

We sincerely thank the Reviewers for their detailed reading of the manuscript, constructive and helpful comments. Below, please find the replies and revisions in response to the Reviewers' comments point-by-point, with the comments of the Reviewers highlighted in blue and italic, followed by our reply and action. Modifications of the manuscript are highlighted in yellow.

Reviewer 1

The authors have responded to the previous review by incorporating additional figures and discussions, and by comparing their work with two other QD condensate studies to assert its superiority. However, I remain concerned about the significance and rigorousness of the work, particularly in light of the critical data on light-matter strong coupling and the presentation and analysis of power dependence. The following points may need to be addressed before considering publication in Nature Communications.

Our response: We thank the Reviewer for acknowledging our improvements and additions in the previous revision, and we will address the remaining criticism point-by-point in the following.

Firstly, the authors should elaborate on the importance of QD film polariton condensate from two key aspects to support the significance of their work. The condensation reported is achieved in an ensemble of QDs rather than at the single or few-QD level, raising questions about its significance compared to numerous studies reporting room temperature polariton condensation in perovskite microcrystals. Does this QD ensemble condensate offer more robust quantum features against dephasing, enabling further exploration of such interacting quantum fluids of light, or does it remain within the classical regime of coherence?

Our response: We thank the Reviewer for their constructive criticism, which has helped us better delineate the novelty of our work. We would like to emphasize that our cavity polariton condensate is actually the first one with **any** QDs, be it halide perovskites or conventional semiconductors (III-V or II-VI). There are several aspects that set apart our work using an ensemble of perovskite QDs from the many previous studies with perovskite microcrystals, opening the door towards further exploration and new polaritonic applications with perovskites:

- 1) The wavelength of QDs can be tuned by changing their size whereas the excitons in microcrystals are fixed to the bulk wavelength.
- 2) The use of chemically synthesized colloidal QDs versus microcrystals (grown via CVD or inverse temperature crystallization) enables very flexible, low-cost fabrication and processing.
- 3) Bulk, 2D or 1D materials have a continuous excitonic density of states in contrast to QDs with their discrete density of states owing to the strong 3D confinement. While our present study does not give conclusive results how the polaritonic interactions are modified by the

confinement and the concomitant discretization of states and energies, future further improvements of the open cavity stability and tailored cavity nanostructures should allow to address this. As QD lasers can have clear advantages over quantum well lasers in conventional lasers, such as better temperature performance, lower threshold, and faster switching speed (see the Reviewer's second comment below and e.g. Coleman et al., <https://doi.org/10.1109/JLT.2010.2098849>) it would be advantageous to use similar effects with polariton condensates with QDs.

4) Entering the quantum regime requires strong polariton-polariton interactions. Since such quasiparticles primarily interact through their excitonic component, enhancing exciton-exciton Coulombic interactions via quantum confinement could be a key enabler. This necessitates the use of QD with size-dependent and strong exciton-exciton interactions (see for example, <https://advanced.onlinelibrary.wiley.com/doi/10.1002/adma.202208354>)

5) As the Reviewer explicitly indicates ("*Does this QD ensemble condensate offers more robust quantum features against dephasing...?*"), reducing dephasing processes is highly relevant to achieve polariton condensation. In this context, QD networks provide appealing advantages due to their reduced exciton-phonon coupling with respect to quantum well structures (e.g., 2D perovskites). Furthermore, exciton-exciton annihilation, well known to be detrimental for condensate formation, is reduced in QD solids, as exciton mobility is reduced with respect to other perovskite nanostructures.

6) We concur with the Reviewer that there are also interesting follow-up questions how quantum fluid properties are altered when using an ensemble of discrete quantum emitters. Therefore, even without going to strong coupling of single/few QDs, our experiments with QD ensembles are likely to stimulate a wide range of new research, and we consider them to be the first but not the "final word" on polariton condensates with QDs.

Our action: We have highlighted some of these aspects in the introduction part of the manuscript.

Compared to these macroscopic single crystals, colloidal nanocrystal QDs have the advantages of wavelength tunability²² and size-dependent, strong exciton-exciton interactions^{42,43}, highly engineerable and flexible synthesis, deposition and processing, and a discrete, non-continuous density of electronic states – a feature that has allowed conventional semiconductor QDs to become superior laser gain materials over their quantum well counter parts⁴⁴ and likely could have far-reaching implications also for the quantum fluid properties of polariton condensates.

Secondly, QDs have been widely utilized as superior gain materials and QD lasers have been realized in VCSEL configurations similar to this work. What distinguishes the achievement of polariton condensation from photonic lasing? Except for the phase coherence the authors demonstrated, the intensity super-linear growth and linewidth narrowing are identical with that of the photon laser. And divergence angle narrowing feature is missing in this work considering the trapped geometry that broadens wavevector distribution of polariton condensate.

Our response: The differences between polariton condensates and photon lasers are discussed extensively in the literature (see e.g. reviews by Byrnes et al., Nature Physics

(2014) <https://doi.org/10.1038/nphys3143>, Bloch et al. Nature Reviews Physics (2022) <https://doi.org/10.1038/s42254-022-00464-0>, and with a focus on applications in information processing Kavokin et al., Nature Review Physics (2022) <https://doi.org/10.1038/s42254-022-00447-1>). These range from fundamental aspects of the condensation process (e.g. the partial thermalization allows to study Kibble-Zurek and Kardar–Parisi–Zhang physics, relevant for phase transitions both in solid state and in astronomical models) to nonlinearities down to single-photon level, even with condensation in a large ensemble of emitters (Zasedatelev et al., Nature (2021), <https://doi.org/10.1038/s41586-021-03866-9>).

The Reviewer is right that our polariton condensate has a broader divergence angle (Fig. 2b) compared to condensates in planar, multimode VCSEL-like configurations. It is important to remember that this is a single mode rather than a distribution of states of various wavevectors. Additionally, it is Fourier-limited by the wavelength-scale lateral confinement from the Gaussian deformation, which permits diffraction-limited focusing. Notably, the divergence angle of our single-mode polariton condensate is comparable to the 20° typically achieved with single-mode VCSELs. In fact, similar to conventional lasers, the single-mode nature of the polariton condensate is pivotally important for many device applications as it allows to circumvent dispersion effects and enables compact devices, something that is not achieved with planar, multimode VCSEL-like cavities. As discussed above, comparing polariton condensation with conventional lasing, however, provides additional benefits and widens the possibilities for both fundamental studies and efficient devices.

Our action: We have added the references to the above mentioned reviews of the differences and benefits of polariton condensates and photon lasers to the introduction.

The differences and beneficial aspects of such polariton condensates compared to photon lasers have been thoroughly explored^{4–6}, and they provide an excellent basis for studying and exploiting quantum fluids of light^{7–9} and building optoelectronic devices that benefit from their nonlinearity and interactions^{10,11}.

Furthermore, we have included a statement on the importance of single-mode condensation to the conclusions.

In the condensate regime, we observed emission in a single, discrete mode – an important aspect for potential device applications – and with an extended temporal coherence reaching 5.2 ps.

Some detailed comments are:

1. The authors have added strong coupling anti-crossing data in the revised Supplementary information. However, in Figure 2, which is the primary data of the work stating the formation of polariton condensate, the polariton branch with quantized energy states and anti-crossing at excitonic energy is not visible. An angle-resolved spectra with broader wavelength distribution (similar to the wavelength range in Extended Figure 8) would be needed to convince the formation of exciton polaritons in the Gaussian-defect cavity, rather than solely with cavity length-dependent measurement in Extended Figure 8. Such measurement is feasible as reported by other open-cavity works (Nature Communications, 12, 4933 (2021)).

Our response: We would like to emphasize an important difference between the strong-coupling anti-crossing data in Fig. 1b, Ext. Data Figs. 5 and 8 and the polariton condensation data in Figs. 2, 3 and Ext. Data Fig. 9. The former data use white-light transmission measurements that probe the density of states over a large wavelength range, in order to reveal the existence of the anti-crossing between the lower and upper polariton branches as well as the manifold of Laguerre-Gaussian states of the Gaussian deformation. In contrast, the latter data use non-resonant excitation, creating an exciton reservoir that subsequently relaxes and populates the polariton states, forming a (partly) thermalized polariton occupation whose emission is then detected. As the detuning of exciton and cavity for our polariton condensates by far exceeds the thermal energy (100 meV vs. 25 meV at 300 K), there is effectively no population of energetically higher states (i.e. extending Fig. 2 towards higher energies would be an empty plot), and hence, the anti-crossing of states cannot be observed with the excitation scheme used for polariton condensation. This is by no means an exception, as a large part of polariton condensates in the literature are created with anti-crossings larger than the respective thermal energy (see review by Sanvitto and Kéna-Cohen, Nature Materials (2016) <https://doi.org/10.1038/nmat4668>).

In order to show that indeed our polaritons are almost thermalized as is required for polariton condensates (versus polariton lasing), we performed a new analysis of the data in Fig. 3, comparing the emission between the LG00 and the LG01 states below threshold. From the intensity ratio I_{LG01}/I_{LG00} and assuming Maxwell-Boltzmann distribution $I_{LG01}/I_{LG00} = \exp(-(E_{LG01} - E_{LG00}) / k_B T_{\text{eff}})$ we obtain an effective polariton temperature $T_{\text{eff}} = (228 \pm 10)$ K, which is close to room temperature but indicates not fully complete thermalization, similar to other trapped polariton condensates [Balili et al., Science (2007), <https://doi.org/10.1126/science.1140990>].

There are two reasons why in Nature Comm. 12, 4933 (2021) (<https://doi.org/10.1038/s41467-021-24925-9>), mentioned by the Reviewer, the angle-resolved emission spectra (Fig. 1b of the reference) show anti-crossing: 1) Both splitting and detuning are smaller than the thermal energy, allowing the photoluminescence to illuminate all the states. 2) The angle-resolved spectra that show the anti-crossing are actually obtained on a flat cavity, and not on the nanostructured part that contains the lattice. Hence, this method is very indirect, as it shows the polariton dispersion not for the cavity modes that are actually studied in the rest of the manuscript (Figs. 2-4 of the reference), but on a different position, which might have different roughness (flat cavity vs. nanostructures) and concomitant losses and different excitonic properties (film thickness, disorder), and therefore different light-matter coupling strength.

Our method (Fig. 1b and Ext. Data Figs. 8), however, where we directly bring the exciton and the relevant wavelength-sized Laguerre-Gaussian cavity modes in-situ into energetic resonance is the most direct and moreover, well-established way to prove strong coupling (see Weisbuch et al., Phys. Rev. Lett. (1992) <https://doi.org/10.1103/PhysRevLett.69.3314> and Khitrova et al., Rev. Mod. Phys. (1999) <https://journals.aps.org/rmp/abstract/10.1103/RevModPhys.71.1591>), allowing us to measure the coupling strength of both trapped Laguerre-Gaussian states and untrapped cavity modes at the same time (60 meV vs. 53 meV). While for flat cavities with their continuous, quasi-parabolic density of states, angle-resolved measurements allow to identify the anti-crossing with angle-tuning, discrete modes as within our Gaussian deformations have no angular

dependence, as seen in Fig. 2, and therefore angular-resolved measurements cannot be used to observe the anti-crossing or thermalization/condensation of polaritons.

Our action: We included the references that the anti-crossing characteristic for strong coupling can be directly measured by bringing the exciton and cavity modes into resonance by tuning the cavity, mentioning moreover that angle-resolved anti-crossing like in planar cavities cannot be observed for the discrete Laguerre-Gaussian states.

In contrast to PC, which allow angle-tuning due to their continuous, quasi-parabolic dispersion, the discrete, dispersion-less nature of the localized LG states necessitates the direct measurement of the anti-crossing characteristic in the strong coupling regime via bringing the exciton and the cavity modes into resonance through tuning of the cavity length^{1,54}.

Furthermore, we have included the calculation of the effective temperature in the manuscript:

From the intensity ratio I_{LG01}/I_{LG00} below threshold and assuming Maxwell-Boltzmann distribution $I_{LG01}/I_{LG00} = \exp(-(E_{LG01}-E_{LG00}) / k_B T_{\text{eff}})$, with k_B being the Boltzmann constant, we obtain an effective polariton temperature $T_{\text{eff}} = (228 \pm 10)$ K, which is close to room temperature but indicates not fully complete thermalization, similar to other trapped polariton condensates⁵⁶.

2.As the primary data showing pump fluence dependent condensation process in the cavity polariton, the data quality in Figure 2 should be improved a lot. At least 3 high quality angle-resolved PL spectra with high signal-to-noise ratio would be needed to demonstrate the evolution from below threshold to above threshold with evident blueshift and linewidth narrowing with intensity counts labelled on each color bar, instead of showing just 2 quantized cavity modes at 2 pump fluences.

Our response: As mentioned above, angle-resolved measurements have no real relevance for individual modes with energetically flat angular dependence as the Laguerre-Gaussian states. Furthermore, as the Reviewer noticed, the spreading in k-space of the emitted photons strongly reduces the signal-to-noise ratio in a systematic way. Therefore, the angular-integrated spectra in Fig. 3 and Ext. Data Fig. 9 contain the same information but with much less noise, allowing to reliably extract blueshift, linewidth and intensity threshold. Fig. 3 and Ext. Data Fig. 9 show representative data from two samples and at two positions each, demonstrating the robustness and the reproducibility of the polariton condensate observation. As a general remark on the signal-to-noise in our study, we like to point out that our wavelength-sized cavities have a modal extension about two orders of magnitude smaller than typical flat cavities with pump spots of 5-10 μm diameter, and therefore, the number of photons is reduced accordingly.

3.The authors claimed that the blueshift does not saturate and its magnitude stays significantly below coupling strength, proving that samples remain in strong coupling regime at high pump fluence. However, in response to last time's concern on interpretation of the large blueshift behavior while possessing small exciton fraction, the authors stated that the blueshift stems from various reasons (excitation-induced changes in the refractive index and partly phase-space filling

effects of the optical transitions, interactions with the exciton reservoir), therefore one can have an appreciable blueshift in the system, even if the polariton has a small exciton fraction. It is challenging to reconcile these two claims. How can the origin of the one part of the blueshift be clearly attributed to polariton interactions, given the significant contributions from other factors? Additional experiments, such as resonant excitation reflectivity measurements (Nature 591, 61–65 (2021)), may be necessary to elucidate the origin of the blueshift. Furthermore, considering the open cavity geometry, it might be feasible to tune the cavity to obtain samples with larger excitonic fractions.

Our response: Indeed, the experimental data shows two aspects that are hard to reconcile:

- 1) The blueshift seems to be similar for both the LG00 (condensed) and the LG01 (uncondensed) state, suggesting that it is not arising from the respective population of the polariton mode, which is very different for LG00 and LG01.
- 2) The blueshift appears to have a similar threshold behavior as the condensate, i.e. it seems to be sensitive of the respective population of the polariton condensate when crossing the threshold and less so with phase-space filling and reduction of the Rabi splitting, as there no threshold behavior would be expected.

Therefore, while we discuss the signatures that we observe in the data, we refrain from a firm attribution and give a fair and honest conclusive statement: “Hence, based on the current experimental data, we cannot fully conclude about potential modifications of the polariton interactions due to the excitonic confinement in the QDs.”

Nevertheless, we agree with the Reviewer that additional further experiments could shed light on the question about strength and origin(s) of the interactions. This is why we had already included such statement on the realization of schemes that circumvent the population of the exciton reservoir in the last paragraph of the manuscript: “Moreover, such wavelength-scale potential arrays and patterned excitation beams can help to determine and observe polariton interactions in the future, ...” Resonant femtosecond reflectivity measurements, as in the manuscript (Nature 591, 61 (2021)) the Reviewer suggested, could be an alternative. Yet, from the decades of polariton research, one can conclude that proper, unambiguous measurements of polariton-polariton interactions are challenging and nontrivial and an effort on their own right, as even for established materials like III-V semiconductors often controversial values are published in the course of the years. Therefore, our current work focusses on the first realization of polariton condensation with quantum dots, leaving further, carefully set up experiments to unambiguously quantify the interactions for future experiments.

Our action: We have added the suggested reference on the resonant reflection measurement to the discussion of future interaction measurements.

Moreover, such wavelength-scale potential arrays and patterned excitation beams, or alternatively, resonant excitation schemes⁶⁶, can help to determine and observe polariton interactions in the future, and ideally push the system into the polariton blockade regime, like it has been achieved for epitaxially grown semiconductor QDs in photonic crystal cavities⁶⁷.

Reviewer 2

I am satisfied with the changes. I still disagree with the condensation terminology for a trapping potential where only 2 modes are relevant - this is very different from the examples given by the authors such as the Science paper or cold atom traps for BEC which are still highly multimode, but this is mostly semantics and abused quite a bit by others as well. Overall I support publication.

Our response: We thank the Reviewer for the positive assessment. Indeed, two modes are the absolute minimum number of discrete states where something like polariton condensation with macroscopic population of the lower state over the higher one can be discussed. In order to show that nevertheless a quasi-thermalized occupation is observed, we performed a new analysis of the data in Fig. 3, comparing the emission between the LG00 and the LG01 states below threshold. From the intensity ratio I_{LG01}/I_{LG00} and assuming Maxwell-Boltzmann distribution $I_{LG01}/I_{LG00} = \exp(-(E_{LG01}-E_{LG00}) / k_B T_{eff})$ we obtain an effective polariton temperature $T_{eff} = (228 \pm 10)$ K, which is close to room temperature but indicates not fully complete thermalization, similar to other trapped polariton condensates [Balili et al., Science (2007), <https://doi.org/10.1126/science.1140990>].

Our action: We have included the calculation of the effective temperature in the manuscript:

From the intensity ratio I_{LG01}/I_{LG00} below threshold and assuming Maxwell-Boltzmann distribution $I_{LG01}/I_{LG00} = \exp(-(E_{LG01}-E_{LG00}) / k_B T_{eff})$, with k_B being the Boltzmann constant, we obtain an effective polariton temperature $T_{eff} = (228 \pm 10)$ K, which is close to room temperature but indicates not fully complete thermalization, similar to other trapped polariton condensates⁵⁶.

Reviewer 3

The authors have addressed all of my queries and recommendations, and the paper's structure now meets the publication standards of Nature Communications.

Our response: We thank the Reviewer for the positive assessment.